# Interleukin 27, Similar to Interferons, Modulates Gene Expression of Tripartite Motif (TRIM) Family Members and Interferes with Mayaro Virus Replication in Human Macrophages

**DOI:** 10.3390/v16060996

**Published:** 2024-06-20

**Authors:** Lady Johana Hernández-Sarmiento, Y. S. Tamayo-Molina, Juan Felipe Valdés-López, Silvio Urcuqui-Inchima

**Affiliations:** Grupo Inmunovirología, Facultad de Medicina, Universidad de Antioquia UdeA, Calle 70 No. 52-21, Medellín 050001, Colombia; ladyj.hernandez@udea.edu.co (L.J.H.-S.); yordi.tamayo@udea.edu.co (Y.S.T.-M.); felipe.valdes@udea.edu.co (J.F.V.-L.)

**Keywords:** tripartite motif, interferons, interleukin 27, macrophages, RNA-seq

## Abstract

Background: The Tripartite motif (TRIM) family includes more than 80 distinct human genes. Their function has been implicated in regulating important cellular processes, including intracellular signaling, transcription, autophagy, and innate immunity. During viral infections, macrophages are key components of innate immunity that produce interferons (IFNs) and IL27. We recently published that IL27 and IFNs induce transcriptional changes in various genes, including those involved in JAK-STAT signaling. Furthermore, IL27 and IFNs share proinflammatory and antiviral pathways in monocyte-derived macrophages (MDMs), resulting in both common and unique expression of inflammatory factors and IFN-stimulated genes (ISGs) encoding antiviral proteins. Interestingly, many TRIM proteins have been recognized as ISGs in recent years. Although it is already very well described that TRIM expression is induced by IFNs, it is not fully understood whether TRIM genes are induced in macrophages by IL27. Therefore, in this study, we examined the effect of stimulation with IL27 and type I, II, and III IFNs on the mRNA expression profiles of TRIM genes in MDMs. Methods: We used bulk RNA-seq to examine the TRIM expression profile of MDMs treated with IFNs or IL27. Initially, we characterized the expression patterns of different TRIM subfamilies using a heatmap. Subsequently, a volcano plot was employed to identify commonly differentially expressed TRIM genes. Additionally, we conducted gene ontology analysis with ClueGO to explore the biological processes of the regulated TRIMs, created a gene-gene interaction network using GeneMANIA, and examined protein-protein interactions with the STRING database. Finally, RNA-seq data was validated using RT-qPCR. Furthermore, the effect of IL27 on Mayaro virus replication was also evaluated. Results: We found that IL27, similar to IFNs, upregulates several TRIM genes’ expression in human macrophages. Specifically, we identified three common TRIM genes (*TRIM19*, *21*, and *22*) induced by IL27 and all types of human IFNs. Additionally, we performed the first report of transcriptional regulation of *TRIM19*, *21*, *22*, and *69* genes in response to IL27. The TRIMs involved a broad range of biological processes, including defense response to viruses, viral life cycle regulation, and negative regulation of viral processes. In addition, we observed a decrease in Mayaro virus replication in MDMs previously treated with IL27. Conclusions: Our results show that IL27, like IFNs, modulates the transcriptional expression of different TRIM-family members involved in the induction of innate immunity and an antiviral response. In addition, the functional analysis demonstrated that, like IFN, IL27 reduced Mayaro virus replication in MDMs. This implies that IL27 and IFNs share many similarities at a functional level. Moreover, identifying distinct TRIM groups and their differential expressions in response to IL27 provides new insights into the regulatory mechanisms underlying the antiviral response in human macrophages.

## 1. Introduction

Macrophages are key components of innate immunity that play an essential role in initiating inflammatory and antiviral immune responses to combat viruses and other pathogens. The activation of various pattern-recognition receptors (PRRs) in macrophages by recognition of different pathogen-associated molecular patterns (PAMPs) leads to the production of multiple pro-inflammatory and antiviral cytokines [1,2], including interferons (IFNs) [3]. Based on the degree of sequence homology, structural features, receptor usage, and biological effects, the IFN family has been divided into three types in humans: Type I IFN (IFN-I) has several subtypes, including IFN alpha (α), beta (β), kappa (ĸ), delta (δ), epsilon (ε), and omega (ω). These subtypes collectively interact with a heterodimeric receptor (IFNAR) consisting of IFNAR1 and IFNAR2 [4,5]. Type II IFN (IFN-II) is represented by IFN gamma (γ) that signals through the IFN gamma receptor complex (IFNGR), consisting of IFNGR1 and IFNGR2 [5,6,7,8]. Type III IFN (IFN-III) includes the IFN lambda (λ)-1-4 subtypes that bind to the heterodimeric IFN lambda receptor (IFNLR), consistent with the IL10RB and IFNLR chains [8,9,10]. IFN-I and IFN-III bind to cell surface receptors, activating intracellular signaling pathways, which include Janus kinase 1 (JAK1), Tyrosine kinase 2 (TYK2) [10,11,12], and signal transducer and activator of transcription (STAT) 1 and STAT2 proteins [13,14]. The JAK1/TYK2/STAT axis, along with IFN regulatory factor 9 (IRF9), form a transcription factor complex known as IFN-stimulated gene factor 3 (ISGF3) [15]. These activated complexes enter the nucleus and induce the expression of a large number of IFN-stimulated genes (ISGs), which orchestrate the antiviral state in cells [8,10,11,16,17,18,19,20] On the other hand, IFN-II/IFNGR interaction results in the activation of JAK1 and JAK2, which phosphorylate and activate STAT1 and, to a lesser extent, STAT3 [21]. Once activated, STAT1 homodimerizes to form the transcriptional regulator IFNG-activated factor (GAF), which binds to the IFNG-activated sequence (GAS) elements in the promoter of IFNG-responsive genes [22], involved in orchestrating a pro-inflammatory response and antiviral state in cells.

Interleukin 27 (IL27) is a heterodimeric cytokine member of the IL6 and/or IL12 family of cytokines, consisting of the protein subunits IL27p28 and Epstein-Barr virus-induced gene 3 (EBI3) [23,24]. Both IL27 subunits have differential transcriptional regulation, and both are highly expressed by antigen-presenting cells (APCs), including macrophages, in response to the activation of Toll-like receptors and transcription factors (TF) such as IRF1 and nuclear factor kappa B (NF-kB), which promote IL27p28 and EBI3 gene expression, respectively [25]. The IL27 signals through heterodimeric IL27 receptor (IL27R) consistent of IL27Rα and gp130, and like IFN-II, triggers the activation of JAK1 and JAK2, which in turn phosphorylates and activates the STAT1 and STAT3 transcription factors [26,27,28,29]. IL27 has various immunomodulatory functions that play a critical role in immune system activity [23], including stimulation of T helper 1 (Th1) immune responses by inducing expression of T-bet TF and IL12Rβ2 in a STAT1-dependent manner and promoting IFNγ production in activated CD4+ T cells [23,30,31,32,33]. On the other hand, IL27 inhibits Th2 and Th17 differentiation by downregulating the expression of GATA binding protein 3 (GATA3) [30,31,33,34] and Retinoid orphan nuclear receptor C gamma T (RORγt) [35,36], respectively. Additionally, IL27 promotes the production of both pro- and anti-inflammatory mediators by activating STAT1, STAT3, and NF-κB TFs [37,38]. We recently published that IL27 and IFNs induce transcriptional changes in various genes, including those involved in JAK-STAT signaling [39]. Furthermore, in this study, we have shown that IL27 and IFNs share proinflammatory and antiviral pathways in monocyte-derived macrophages (MDMs), resulting in both common and unique expression of inflammatory factors and IFN-stimulated genes (ISGs) encoding antiviral proteins. Moreover, IL27 has gained significant relevance since it promotes antiviral activity. For instance, IL27 elicits a robust antiviral response against different viruses, including human immunodeficiency virus type 1 (HIV-1) [40], hepatitis B virus (HBV) [41], hepatitis C virus (HCV) [40], influenza A virus (IAV) [42], zika virus (ZIKV) [29,43], and chikungunya virus (CHIKV) [44].

The Tripartite motif (TRIM) family includes more than 80 distinct human gene members. They are characterized by the presence of an N-terminal tripartite motif formed by a RING-finger (R) domain, one or two B box (B) domains, and a coiled-coil region, known as RING-B-box-Coiled-Coil (CC) domains. The CC domain is known to be involved in promoting homo- and heteromeric interactions, resulting in the formation of oligomers and complexes of higher-order structures [45]. The RBCC domain of each TRIM is followed by a variable C-terminal domain. TRIM proteins are further classified into twelve subfamilies (C-I to C-XI and NO RING) based on differences in the C-terminal composition [46,47]. TRIMs are considered E3 ubiquitin ligases due to the general function of the RING domain, which is known to have E3 ubiquitin ligase activities [48,49]. TRIMs play significant regulatory roles in important cellular processes such as innate immune response, intracellular signaling, transcription, and autophagy via different pathways [50,51,52,53]. Furthermore, TRIMs are involved in the induction of an antiviral state and the control of viral replication in cells [50,51,52]. Several TRIMs show tissue-specific and/or stimulus-specific expression (reviewed in: [54]), and dysregulation of TRIM gene expression could result in diverse pathological conditions, including developmental disorders, autoimmune diseases, and tumorigenesis [55,56].

Different reports show that the transcriptional expression of many TRIM superfamily members is regulated in response to IFNs [54,57,58]. Additionally, accumulating evidence has demonstrated their significant role in restricting viral infections [59,60,61]. Performing a systematic analysis of TRIM gene expressions in MDMs stimulated with IFNs, Carthagena et al. (2009) identified that 27 out of the 72 human TRIM genes are sensitive to IFN stimulation [62]. Rajsbaum et al. (2008) also reported that genes encoding a subset of TRIM proteins located on chromosome 7 were induced by IFN-I in mouse macrophages [63]. However, it remains unclear whether TRIM genes are uniformly induced in macrophages by IL27, similar to what occurs with all three types of IFNs. Therefore, we performed a bulk RNA-sequencing (RNA-seq) approach to examine the effects of stimulation with IL27, IFN-I (α and ε), IFN-II (γ), or IFN-III (λ) on the mRNA levels of TRIMs genes in primary human MDMs. We confirmed the RNA-seq data of selected TRIMs genes by RT-qPCR and assessed the effect of IL27 on Mayaro virus replication.

## 2. Materials and Methods

### 2.1. Ethics Statement

The protocols for enrolling participants and collecting samples were approved by the Bioethics Research Committee of the Universidad de Antioquia’s/Sede de Investigación Universitaria (Medellín, Colombia; approval no 19-08-836; 15 May 2019). Participants provided written informed consent before inclusion, following the principles expressed in the Declaration of Helsinki. This study involved 3 or 4 healthy donors, depending on the experiment, without technical triplicates.

### 2.2. Transcriptomic Analysis of Previously Published RNA-Seq Studies and Prediction of TRIM Expression in MDMs

To explore the expression of TRIMs in MDMs treated with IFNα, IFNε, IFNγ, IFNλ, and IL27, we re-analyzed two RNA-seq datasets. The first RNA-seq dataset consisted of MDMs treated with IFNs, was downloaded from the Gene Expression Omnibus (GEO) database under accession number GSE158434, and was analyzed following the workflow described in Figure 1. This was obtained from MDMs, either untreated or treated with 25 ng/mL of IFNα, IFNε, IFNγ, or IFNλ (n = 3) for 18 h. The second RNA-seq dataset (accession number GSE262963) was generated from our own research on MDMs treated with 25 ng/mL of IL27 (n = 3) for 18 h. Untreated cells were used as a control.

### 2.3. Data Annotation and Batch Effect Correction

The raw counts from GSE158434 and RNA-seq data for MDMs treated or untreated with recombinant human IL27 were processed using a workflow executed in R software (version 4.2.0) [64]. Based on the genotype, we selected only protein-coding genes. Batch effect correction on raw count data for each transcriptome was performed using ComBat-seq [65]. The essential information regarding the TRIMs included in the analysis is detailed in Appendix A [66,67,68].

### 2.4. Analysis of TRIM Gene Expression

To identify the top differentially expressed TRIM genes (DETGs), we utilized the DESeq2 library [69] and calculated the Log_2_Fold Change (FC) (Treatment/Control). Subsequently, we plotted a heatmap of the FC for TRIM genes using the pheatmap library and generated volcano plots using the ggplot2 library in R [64]. The criteria applied for the volcano plots were a false discovery rate (FDR) < 0.05 and a FC > 0.6 [25,43,44]. Finally, we used Cytoscape 3.9.1 software [70] to visually represent the common genes among the transcriptomes of MDMs treated with IFNα, IFNε, IFNγ, IFNλ, and IL27.

### 2.5. Gene Set Enrichment Analysis

Enrichment analysis was performed using the ClueGO plugin (version 2.5.9) [71] with default parameters and GO term fusion, applying a significance threshold of *p* ≤ 0.05. We used GeneMANIA, an online platform, to analyze TRIM genes and their co-expressed gene network. The results indicated the biological terms and pathways in which the genes are enriched.

Gene’s networks, consisting of 20 interactors and exhibiting high confidence (*TRIM19*, *TRIM21*, *TRIM24*, *TRIM32*, and *TRIM58*) as well as medium confidence (*TRIM22* and *TRIM69*), were performed using the STRING database v11.5 [72]. The networks were downloaded and merged in Cytoscape (version 3.9.1) for a proper representation [70]. Subsequently, functional enrichment analysis was conducted using the STRING plugin in Cytoscape.

### 2.6. Culture of Primary Human Monocytes and Differentiation into Monocyte-Derived Macrophages

Human peripheral blood mononuclear cells (PBMCs) were isolated from leukocyte-enriched blood units obtained from healthy donors through density gradient separation using Lymphoprep (STEMCELL Technologies Inc., Vancouver, BC, Canada) by centrifugation at 850× *g* for 21 min. For platelet depletion, the cells were washed three times with 1X PBS (Sigma-Aldrich, St. Louis, MO, USA) at 250× *g* for 10 min each round. The percentage of CD14-positive cells was then determined using flow cytometry. To obtain human monocytes, 24-well plastic plates were scratched with a 1000 μL pipette tip and seeded with 5 × 10^5^ CD14 positive cells per well to allow their adherence during 2 h in RPMI-1640 medium (Sigma-Aldrich) supplemented with 0.5% autologous serum, 4 mM L-glutamine, and 0.3% NaCO_3_ and cultured at 37 °C with 5% CO_2_. Non-adherent cells were removed by washing twice with 1X PBS and monocytes were cultured in RPMI-1640 medium supplemented with 10% FBS, 4 mM L-glutamine, 0.3% NaCO_3_, and 1% antibiotic-antimycotic solution 100X (complete medium) and incubated at 37 °C and 5% CO_2_ for 6 days to obtain MDMs, as previously described [73].

### 2.7. Treatment of Monocyte-Derived Macrophages with IL27, IFNβ, IFNγ, or IFNλ1

MDMs were treated independently with 1 UI/mL of recombinant human IL27 (BioLegend, San Diego, CA, USA), IFNβ (STEMCELL Technologies Inc.), IFNε, or IFNλ1 (BioLegend). Cells were harvested at 18 h and stored at −80 °C to quantify TRIMs’ mRNA expression by RT-qPCR.

### 2.8. Cell Lines and Mayaro Virus Stocks

We used a Brazilian clinical isolate of MAYV kindly gifted by Professor Mauricio Nogueira (Faculdade de Medicina de São José do Rio Preto, São José do Rio, SP, Brazil). As previously reported [74], MAYV was propagated in Vero cells (ATTC CCL-81).

### 2.9. In Vitro MAYV Infection of MDM and In Vitro Antiviral Assay

Human MDMs were pre-treated with increasing concentrations (1, 5, 10, and 25 ng/mL) of recombinant human IL-27 or 25 ng/mL IFN-β1. After 6 h pre-treatment, MDMs were infected with MAYV at MOI 05 at 37 °C with 5% CO_2_ for 1.5 h, as previously reported. Briefly, the cells were washed with PBS-1X to remove the unbound virus, and a fresh complete medium was added and incubated at 37 °C with 5% CO_2_. Both supernatants and cells were harvested at 24 h post-infection (h.p.i). Cells were lysed, and RNA was used for RT-qPCR. Viral replication of MAYV was evaluated by a plaque assay on Vero cells.

### 2.10. RNA Extraction, cDNA Synthesis, and Real-Time RT-qPCR

Total RNA was extracted from MDMs, whether treated or not with IL27, IFNβ, IFNγ, or IFNλ1, as well as from MDMs that were either uninfected or infected with Mayaro virus, using TRIzol reagent (Invitrogen, Life Technologies, Carlsbad, CA, USA) following the manufacturer’s instructions. The concentration was determined using a NanoDrop-1000 spectrophotometer (Thermo Scientific, Wilmington, DE, USA). To synthesize copy DNA (cDNA) from RNA, the commercial iScript™ cDNA Synthesis Kit (Bio-Rad, Hercules, CA, USA) was used following the manufacturer’s instructions. Real-time RT-qPCR amplifications were carried out using the SsoAdvanced TM Universal SYBR^®^ Green Supermax (Bio-Rad, USA), and primer sequences were *TRIM19*, forward, 5′-CCGTCATAGGAAGTGAGGTCTTC-3′, and reverse, 5′-GTTTTCGGCATCTGAGTCTTCCG-3′. *TRIM21*, forward, 5′-CAGAACTCAGGAGTGTGTGCCA-3′, and reverse, 5′-TCCAAGCCTCACTTGTCTCCGA-3′. *TRIM22*, forward, 5′-GGATCGTCAGTAGAGATGCTGC-3′, and reverse, 5′-GAACTTGCAGCATCCCACTCAG-3′. *TRIM25*, forward, 5′-AAAGCCACCAGCTCACATCCGA-3′, and reverse, 5′-GCGGTGTTGTAGTCCAGGATGA-3′. *TRIM31*, forward, 5′-GAGCAGATCCAAGTCTTGCAGC-3′, and reverse, 5′-CTCCTCTAGGACTTGATGCAGG-3′. *TRIM69*, forward, 5′-GGAGCAATGTCTCTTAGCCAAGG-3′, and reverse, 5′-TCTCTGGTTGCCAGCACCTTCA-3′. *GAPDH*, forward, 5′-TCGGAGTCAACGGATTTGGT-3′, and reverse, 5′-TGAAGGGGTCATTGATGGC-3′. The Bio-Rad CFX manager was used to obtain the cycle thresholds (Ct) determined for each sample using a regression fit in the linear phase of the PCR amplification curve. Relative mRNA expression of each target gene was normalized to the uninfected control and housekeeping gene GAPDH, using the ΔΔCt method, and |Log_2_Fold Change| > 0.6 was used as the threshold to determine the significant difference in gene expression. We used n = 4 individuals without technical triplicates.

## 3. Results

### 3.1. Identification of TRIM mRNA Expression in MDMs Treated with IL27 or IFNs

To know and compare the TRIM expression profile induced by IL27 and the three types of IFNs, we used bulk RNA-seq to analyze the mRNA expression levels in human MDMs following stimulation with 1 UI/mL recombinant human IL27, recombinant human type I (IFNα and ε), type II (IFNγ), or type III (IFNλ) for 18 h. To define the differentially expressed TRIM genes (DETG), we selected TRIM genes with an FDR < 0.05 and |Log_2_FC (IFN or IL27-stimulated MDMs/Unstimulated MDMs)| > 0.6 [25,43,44]. Then, using a heatmap for the TRIM genes expression profile, the TRIMs were clustered into two groups based on their expression patterns in response to different treatments: the first cluster includes TRIM gene that were induced by IFN-I (IFNα and IFNε) and IFN-III (IFNλ) (STAT1/STAT2/IRF9 dependent). The second cluster includes TRIM genes induced by IFN-II (IFNγ) and IL27 (STAT1 and STAT3 dependent). The results show that there are at least 72 TRIM genes modulated or not based on the observed changes in our transcriptomic analysis. These include members of the C-I to C-XI subfamilies and seven TRIM lacking RING domains, based on their C-terminal domain organization (Figure 2A). We found that only 45 TRIM genes were up- or downregulated, depending on the treatment, while the expression of 27 TRIM genes was not altered. Of note, the transcriptional expression of *TRIM19* (*PML*), *21*, *22*, and *31* were upregulated, while *TRIM71* was downregulated by all three types of IFNs and IL27 (Figure 2A). Additionally, mRNA levels of *TRIM25*, *26*, and *56* were upregulated by all three types of IFNs but not by IL27, while *TRIM29* and *50* transcriptions were strongly downregulated by IL27. *TRIM14*, which lacks the RING domain, was the only one upregulated in response to IFNε, IFNα, and IFNλ treatment (STAT1/STAT2/IRF9 dependent). Furthermore, results show that the C-IV and C-V TRIM subfamilies were more likely to be induced by IFNs and IL27, while the C-VIII TRIM subfamily appears to be unresponsive or downregulated by these treatments (Figure 2A). In summary, TRIM subfamily members of C-I, C-II, C-IV, C-V, C-VI, and C-VII were the ones most modulated by IL27, as well as all three types of IFNs. In contrast, the transcription level of members of subfamilies C-III, C-VIII, C-IX, and C-XI was not altered by any of the treatments (Figure 2A), suggesting that their transcriptional regulation is independent of IFNs or IL27. This selective regulation of TRIM genes may reflect their specific roles in innate immunity and inflammatory responses.

### 3.2. Differentially Expressed TRIM Genes in MDMs Treated with IL27 or IFNs

After determining the DETGs in MDMs for each condition and plotting their expression using volcano plots, we identified common DETGs grouped using Cytoscape software (https://cytoscape.org/, accessed on 3 January 2023). The analysis identified 24 DETGs in IFNε-treated MDMs, compared to the unstimulated MDMs (Figure 2B). Among these 24 DETGs, 20 were significantly upregulated (*TRIM1*, *6*, *8*, *9*, *14*, *16*, *19*, *21*, *22*, *25*, *26*, *31*, *33*, *36*, *38*, *41*, *56*, *62*, *68*, and *69*). On the other hand, the four remaining DEGs were downregulated (*TRIM24*, *32*, *45*, and *54*) (Figure 2B).

In IL27-treated MDMs, only *TRIM19*, *21*, *22*, and *69* were differentially upregulated, and none of the other TRIM genes were significantly downregulated (Figure 2F). Interestingly, expression of these TRIM genes was not previously associated with IL27. To our knowledge and based on the literature, we are the first to report the expression of these TRIMs in response to treatment with IL27.

In IFNα-treated MDMs, the analysis revealed 17 DETGs, out of which 12 were significantly upregulated (*TRIM1*, *6*, *14*, *19*, *21*, *22*, *25*, *26*, *31*, *38*, *56*, and *69*), and 5 remaining DETGs were downregulated (*TRIM9*, *24*, *28*, *32*, and *58*) (Figure 2C). These results suggest that different subtypes of IFN-I induce a differential expression profile of TRIM genes in human MDMs.

In IFNλ-treated MDMs, 11 TRIM genes were found to be differentially expressed, out of which 10 were significantly upregulated (*TRIM1*, *6*, *14*, *19*, *21*, *22*, *25*, *26*, *31*, and *56*), while *TRIM58* was downregulated (Figure 2D).

As shown in Figure 2E, in IFNγ-treated MDMs, the analysis revealed 20 TRIM genes differentially expressed, with 18 being significantly up-regulated (*TRIM1*, *2*, *7*, *9*, *10*, *15*, *19*, *21*, *22*, *25*, *26*, *31*, *38*, *41*, *52*, *63*, and *69*), while *TRIM32 and 54* were downregulated (Figure 2E).

Analysis of common DETGs revealed 11 different clusters of TRIM genes dependent on specific stimuli (Figure 2G). Cluster 1 includes upregulated expression of *TRIM19*, *21*, and *22* (common TRIMs), which were induced by IL27, IFNα, ε, γ, and λ. Cluster 2 comprises *TRIM1*, *26*, *25*, *31*, and *56*, all upregulated only by the three types of IFNs but not by IL27. Cluster 3 includes *TRIM69*, which was induced in response to IL27, IFNα, ε, and γ. Cluster 4 consists of *TRIM9* and *38*, which were upregulated, and *TRIM32*, which was downregulated by IFNα, ε, and γ stimuli. Cluster 5 contains upregulated expression of *TRIM6* and *14* induced by IFN-I (α and ε) and IFN-III (λ). Cluster 6 includes TRIM genes that are IFNε-dependent and comprise *TRIM8*, *16*, *33*, *36*, *62*, and *68*, which were upregulated, while *TRIM45* was downregulated. Cluster 7 includes *TRIM2*, *7*, *10*, *15*, *52*, and *63*, all of which were specifically upregulated by IFNγ treatment. Cluster 8 includes TRIM genes that are IFNε- and γ-dependent, comprising *TRIM41* (upregulated) and *TRIM54* (downregulated). Cluster 9 includes *TRIM58*, downregulated by IFNα and λ. Cluster 10 contains *TRIM24*, downregulated by IFNα and IFNε, while cluster 11 includes *TRIM28*, downregulated by IFNα.

In summary, DETG analysis showed that both IL27 and IFNs broadly activated the expression of TRIM genes. Identifying different TRIM clusters and their differential expression in response to various stimuli provides insight into the regulatory mechanisms underlying the innate immune response and may help develop therapeutic interventions for treating infectious diseases.

### 3.3. Functional Response of TRIM Genes in MDMs Treated with IL27 or IFNs

We used Gene Ontology (GO) to classify the functions of the 45 differentially expressed TRIM genes. Since all three types of IFNs and IL27 influenced TRIM mRNA levels in MDMs, we focused on DETGs upregulated or downregulated, whose expression was dependent on either IL27 and/or IFNs. As shown in Figure 3A, DETGs were enriched in 23 biological pathways. Figure 3B presents the top 9 functional analyses of DETGs along with their respective TRIM genes. Functional analysis of DETGs showed that TRIM genes are enriched in various functions, including regulation of viral processes, positive regulation of DNA-binding TF, defense response to the virus, protein sumoylation, biological processes (BP) involved in interaction with the host, ubiquitination processes, positive regulation of NF-κB signaling, positive regulation of cytokine-mediated signaling, and transcription coactivator activity.

Similarly, we determined the GO network of TRIM genes whose expression was modulated in response to IFNs and/or IL27 treatment (Figure 3C). Functional analysis of DETGs revealed the enrichment of six biological processes related to the ubiquitin process, autophagy, and viral processes. These categories included regulation of the viral entry into host cells, suppression of viral release by the host, negative regulation of viral process, and negative regulation of viral transcription (Figure 3D). Together, results suggest that all types of IFN and IL27 modulate different cellular processes involved in the induction of an antiviral response and the control of different steps of viral replication through the modulation of different TRIM genes in human MDMs.

### 3.4. Gene-Gene Interaction Network of TRIMs Modulated by IL27 and/or IFNs

Transcriptome sequencing showed that expression levels of *TRIM19* (*PML*), *21*, *22*, *25*, *31*, and *69* were significantly up-regulated, whereas *TRIM24*, *32*, and *58* were significantly down-regulated in response to treatment with all types of IFNs and/or IL27. These TRIM genes were analyzed in the GeneMANIA database to find correlated genes based on physical interaction, co-expression, prediction, co-localization, genetic interaction, pathway, and shared protein domains. The gene-gene interaction network revealed that *TRIM21* and *22* genes were significantly enriched in type I IFN production, ubiquitin-protein transferase activity, DNA synthesis involved in DNA repair, regulation of IκB kinase/NF-κB signaling, and regulation of protein deubiquitination (Figure 4A,B). The gene-gene interaction network revealed that *TRIM25* was enriched significantly in multiple functions, including ubiquitin-like protein ligase binding, pattern recognition receptor (PRR) signaling pathway, regulation of IκB/NF-κB signaling, cellular response to virus, regulation of type I IFN production, response to virus, and C-terminal protein amino acid modification (Figure 4C). The gene-gene interaction network showed that *TRIM69* was associated with membrane microdomain, heterotypic cell-cell adhesion, positive regulation of cell-cell adhesion, and extracellular matrix binding (Figure 4D). The gene-gene interaction network showed that *TRIM19* (*PML*) was enriched in cell aging, ubiquitin-like protein ligase binding, response to IFNγ, extrinsic apoptotic signaling pathway, and regulation of cytokine stimulus (Figure 4E). Finally, the gene-gene interaction network showed that *TRIM31* was significantly enriched only in ubiquitin-protein transferase activity and neutrophil receptor binding (Figure 4F).

Regarding the TRIM genes that were significantly downregulated, the gene-gene interaction network revealed that *TRIM58* was significantly associated with the dynein complex (Figure 5A), highlighting its role in this cellular process. Similarly, *TRIM24* was enriched in response to IL7 and IL15, negative regulation of gene silencing by miRNA, and the transcription regulator complex (Figure 5B), indicating its involvement in these processes. *TRIM32* was associated with protein polyubiquitination, response to hypoxia, ubiquitin-protein transferase activity, stress-activated MAPK cascade, and vesicle localization (Figure 5C), further emphasizing its role in these cellular processes. These results suggest that the modulation of specific TRIM genes by IL27 and IFNs induces the modulation of different cellular processes, including antiviral response, cell signaling, cell-cell interaction, programmed cell death, transcriptional regulation, and protein ubiquitination in human MDMs, providing a comprehensive understanding of their impact.

### 3.5. Protein-Protein Interaction Network of TRIMs Modulated by Il27 and/or IFNs

For constructing the protein-protein interaction (PPI) network, we utilized the STRING database v11.5 [72], a widely used online software. This allowed us to construct a comprehensive PPI network of differentially expressed TRIMs modulated by IFNs and/or IL27. The network was then visualized using Cytoscape software. The analysis included the three common TRIMs (*TRIM19* (*PML*), *21*, and *22*), whose expression was modulated by treatment with both IL27 and IFNs. We include, furthermore, those downregulated by IFNα, ε (*TRIM24*), by IFNα, ε and γ (*TRIM32*), and by IFNα and λ (*TRIM58*), as well as those upregulated by IFNα, ε, γ, λ (*TRIM25*, *31* and *69*).

The protein-protein interaction of TRIM21 showed 21 nodes with 65 edges (Figure 6A). These nodes were considered key proteins in the whole network, and most of them were associated with regulation of gene expression, viral processes, cytokine-mediated signaling, positive regulation of protein phosphorylation, PRR signaling, antigen processing, TNFR1-induced NF-ĸB signaling, response to stress, regulation of apoptotic processes, and regulation of IĸB/NF-ĸB signaling.

The TRIM22 protein-protein interaction network showed 21 nodes with 149 edges (Figure 6B). These proteins were linked to antiviral defense, viral processes, IFNα/β signaling and ISG15 conjugation, IFNα/β signaling, and negative regulation of DDX58/IFIH1 (RIG-I/MDA5) signaling, cell surface receptor signaling, cytokine-mediated signaling, and response to IFNα, β, and ɣ. Similarly, TRIM25 was associated with proteins involved in the regulation of gene expression, IĸB/NF-ĸB signaling, regulation of IFNβ and IFN-III, response to IFNα, response to stress, viral process, cytokine-mediated signaling, protein ubiquitination, and PRR signaling (Figure 6C).

Based on the STRING database, we identified 21 nodes and 121 edges in the network associated with TRIM19. The network is linked to various biological processes, including the regulation of apoptotic processes, cell surface receptor signaling, positive regulation of gene expression, cytokine-mediated signaling, viral processes, responses to stress, SUMO E3 ligases, SUMOylate target proteins, PML bodies, IFNγ-mediated signaling, and regulation of IκB/NF-κB signaling (Figure 6D).

In the case of TRIM31, the protein-protein interaction network showed 21 nodes with 38 edges associated with innate immune response, protein ubiquitination, metal ion binding, carnitine synthesis, and protein modification by small protein conjugation or removal (Figure 6E). Interestingly, TRIM31 interacted with other TRIMs, including TRIM14, 29, 33, and 44.

For the downregulated TRIM24, the protein-protein interaction network shows 21 nodes with 87 edges related to several functions, including response to stress, cytokine-mediated signaling, IFNγ-mediated signaling, signaling by BRAF and RAF fusions, SUMO E3 ligases, SUMOylate target proteins, cell surface receptor signaling, PI3K-Akt signaling, protein autophosphorylation, nucleosome, and primary hyperoxaluria type 1, and regulation of immune system processes (Figure 6F).

For TRIM32, the protein-protein interaction network shows 21 nodes with 38 edges involved in regulating cell communication, IFNβ production, antigen processing, stress response, negative regulators of DDX58/IFIH1 (RIG-I/MDA5) signaling, PRR signaling, catabolic process, IL1-induced activation of NF-κB, ubiquitin protein ligase binding, and regulation of IκB/NF-κB signaling (Figure 6G).

Of note, *TRIM69*, which was upregulated in response to IFNα, ε, γ, and IL27, and *TRIM58*, which was downregulated under treatment with IFNα and λ, were not associated with any biological processes (Appendix A, respectively).

### 3.6. Validation of Key TRIMs by RT-qPCR

To further validate the transcriptome sequencing results, six DETGs were selected for RT-qPCR analysis. Among them, we included *TRIM19, TRIM21*, and *TRIM22*, which were induced by IL27 as well as all three types of human IFNs; *TRIM25*, which was induced only by IFNs but not by IL27; and *TRIM69*, which was induced only by IFN-I, IFN-II, and IL27 but not by IFN-III. For this purpose, MDMs were treated with recombinant human IL27 or IFNβ, IFNε, IFNγ, or IFNλ1, or recombinant human IL27, and the cells were harvested at 18 h post-treatment. Consistent with the results of RNA-Seq datasets (Figure 7A), most of the six TRIM genes exhibited similar mRNA expression levels to what was observed in the transcriptomic analysis, mainly for IFN-dependent TRIM genes (*TRIM21*, *22*, *25*, *19*, and *31*; Figure 7B). Furthermore, a significant correlation between RNA-Seq and RT-qPCR for *TRIM19* (R = 0.9971; *p* = 0.001437), *TRIM21* (R = 0.8652; *p* = 0.069821), *TRIM22* (R = 0.9019; *p* = 0.050307), *TRIM25* (R = 0.9983; *p* = 0.000838), *TRIM31* (R = 0.8628; *p* = 0.071127), and *TRIM69* (R = 0.9905; *p* = 0.004775) was observed (Figure 7C), suggesting a strong concordance between bulk RNA-Seq data and RT-qPCR results.

### 3.7. IL27, Similar to IFNs, Promotes the Expression of TRIMs and Interferes with the Replication of MAYV in MDMs

To determine the effect of IL27 stimulation of MDMs on viral replication, MDM cultures were stimulated with increasing concentrations of IL27 or 25 ng/mL IFN-β. Subsequently, cells were infected with MAYV, and viral replication was evaluated at 24 h.p.i by plaque assay on Vero cells. As previously reported [74], MDMs were susceptible to MAYV infection and released many infectious viral particles at 24 h.p.i (Figure 8A). To highlight, the treatment of MDMs with IL27 significantly decreased the release of infectious viral particles in a concentration-dependent manner compared to MAYV-infected control cells, indicating a decrease in MAYV replication. Notably, IFNs induced a higher control over MAYV replication than IL27 (Figure 8A). To obtain a detailed insight into the efficacy of IL27 against MAYV, we next determined the dose-response curve. We calculated the EC50 value, which reflects the concentration of IL27 required to decrease infectious virus particle production by 50%. As observed in Figure 8B, the EC50 value for IL27 was 2.870 ng/mL following infection at MOI 0.5. We next evaluated if the treatment with IL27 and IFN-β induces the expression of *TRIM19*, *21*, *22*, and *69* in MAYV-infected MDM (Figure 8C–F). To this end, based on the plaque assay and EC50 results, we treated the MDM with 10 ng/mL of IL27 or 25 ng/mL of IFN-β. Subsequently, the cells were infected with MAYV, as described above. We observed higher mRNA levels of TRIMs in response to IL27 treatment, with *TRIM22* (Figure 8E), *19* (Figure 8C), *21* (Figure 8D), and *69* (Figure 8F) showing increased expression compared with IFN-β. Notably, the levels of *TRIM 69*, *22*, and *21* were below the threshold we consider significant following IFN treatment and MAYV infection. Furthermore, MAYV infection also promotes the expression of the four TRIMs evaluated at levels very similar to those obtained with IL27 (Figure 8C–F).

## 4. Discussion

Induction of TRIM expression, belonging to the ubiquitously expressed E3 ligase and within the large TRIM protein family, has been extensively studied [50,51,75]. Over the last 20 years, several studies have focused on characterizing the transcriptional profile of TRIM members in response to type I and II IFNs, highlighting their relevance for antiviral activity [62,63]. Unfortunately, IFN-III was not included in these studies. As Carthagena et al., state, IFN III induces antiviral activity using the same IFN-I signaling pathway, the same ISREs, and leads to the induction of almost the same ISGs [62]. Furthermore, the gene expression of these TRIMs has been carried out using microarrays but not RNA-seq.

Several studies have reported the induction of TRIM superfamily members in response to IFNs (reviewed by [50,51], as well, various approaches have been attempted to identify IFN-induced TRIM gene expression [62,63]. However, the precise role of IL27 in promoting the transcription of TRIM genes has not been described. In this study, we used our own bulk RNA-Seq data obtained from MDMs treated with IL27 and public datasets obtained from MDMs treated with each of three types of IFNs to explore the expression profile of all TRIMs.

From our transcriptomic analysis, we found that among the 72 TRIM genes, 45 were either upregulated or suppressed, while the expression of 27 TRIM genes remained unaltered in response to IL27 or IFNs. Interestingly, as was previously reported [76], most up-regulated TRIM genes belong to the C-IV subfamily (*TRIM5*, *6*, *10*, *15*, *21*, *22*, *25*, *34*, *38*, *43*, *48*, *49*, *50*, *60*, *64*, and *69*), while few of them are classified into the C-V subfamily (*TRIM19*, *31*, *40*, *56*, and *61*), the C-III subfamily (*TRIM42*), or the NO RING subfamily (*TRIM14*, *20*, and *29*). Notably, most of these TRIM genes, whether they present the SPRY domain or not, exhibit features that confer protein-protein interactions and RNA binding [77]. C-III, C-VIII, C-IX, and C-XI subfamilies were not altered for any treatments. Among the TRIM genes analyzed, *TRIM14*, lacking the RING domain, was upregulated in response to IFNα, ε, and λ (STAT1/STAT2/IRF9-dependent). Additionally, IFNα, ε, λ, γ, and IL27 upregulated 12, 20, 10, 18, and 4 TRIM genes, respectively (Table 1). Thus, our transcriptomics analysis revealed that both IL27 and IFNs induce the expression of three common genes (*TRIM19*, *21*, and *22*). In addition, IL27, IFN-I, and IFN-II also induce the expression of *TRIM69*, and *TRIM6 and 14* were the only ones induced in response to treatment with type I and III interferon. To highlight, the three types of IFN only promoted the expression of *TRIM1*, *25*, *26*, *31*, and *56*. Of note, for the first time, our study characterized the expression profile of TRIMs genes following IL27 or IFN-III treatment. In contrast, IFNα, ε, λ, and γ suppressed 5, 4, 1 (*TRIM58*), and 2 (*TRIM32* and *54*) TRIM genes, respectively. These results demonstrate that although IFN-I shares the same signaling pathway and ISREs as IFN-III, it does not induce the expression of the same number of ISGs (Figure 2B–D). In summary, the results showed that almost 45 members of the TRIM superfamily are ISGs upregulated or downregulated in response to multiple extracellular stimuli, including IFNs and/or IL27. Furthermore, results show that, like IFNs, IL27 can modulate the transcriptional expression of TRIMs that function as ISGs. These TRIMs are involved in the induction of innate immune and antiviral responses. Together, results suggest that IFNs and IL27 share many similarities at a functional level. Accumulating studies have reported that binding IFNs or IL27 with their respective receptors can activate a subset of downstream signaling pathways, including JAK/STAT-signal pathways, resulting in gene transcription within the target cells to exert host defense function. Furthermore, IL-27-mediated antiviral protein induction is through STAT1- and IRF3-dependent but STAT2-independent manner. Consequently, IFNs and IL27 play essential roles in mediating innate immunity against viral infection through activating JAK-STAT signaling and inducing ISGs. Similarly, we recently published that IFNs and IL27 induce transcriptional changes in various genes, including those involved in JAK-STAT signaling, and lead to a significant upregulation of ISGs associated with innate immune antiviral and pro-inflammatory responses. However, IFNs appear to be more potent than IL-27. Interestingly, an increasing number of TRIM proteins are being recognized as potential viral restriction factors, or ISGs [78,79].

Previous studies in human primary lymphocytes and MDMs have reported that 27 out of 72 TRIM genes are sensitive to both type I and II IFNs [62] (Table 1). The methods used or stimulation time could explain this discrepancy in the number of TRIM genes expressed. In this work, we used bulk RNA-seq, while Carthagena et al. (2009) used RT-qPCR to quantify the expression of the 72 TRIMs. In human monocytes and macrophages stimulated with IFNγ and lipopolysaccharide, a TLR4 ligand, mRNA expression of 26 out of 44 examined TRIM genes was upregulated [80]. In mouse primary macrophages, dendritic cells, and T cells, it was reported that 17 out of 28 TRIMs were increased following influenza virus infection or by TLR signaling in a type I IFN-dependent manner [63]. In the studies reported by Hu et al., (2022), 17 TRIM genes were significantly upregulated in HBV-Associated Hepatocellular Carcinoma [81].

We identified 11 clusters of TRIM genes based on their mRNA levels in MDMs upon stimulation with IL27 or IFNs. We found a cluster of TRIMs, including TRIM1, 25, 26, 31, and 56, whose mRNA expression was upregulated in response to all three types of IFNs but not by IL27 (Figure 2G). We also identified a common set of three TRIM genes, *TRIM19/PML*, *21*, and *22*, that were significantly increased in response to all three types of IFNs and IL27 (Figure 2G). In the human fibrosarcoma cell line HT1080, *TRIM19/PML* and *TRIM21* were found to be induced by both type I and II IFNs, while *TRIM22* (Staf50) expression was up-regulated explicitly by IFN-I [82]. Interestingly, among these TRIMs, *TRIM19/PML*, *21*, *25*, and *26* were previously reported as highly expressed in macrophages and dendritic cells infected with influenza virus in an IFN-I-dependent manner [63]. Furthermore, *TRIM6* and *14* exhibited upregulation in response to type I and III IFNs (STAT1/STAT2/IRF9-IFN-dependent). These TRIMs demonstrated high expression levels in macrophages upon IFN-I stimulation [62] and during influenza virus infection in an IFN-I-dependent manner [63]. While previous reports (Ref. [62], indicated that certain TRIM genes are induced by IFNs, our present study shows, for the first time, that IL27 also induces the expression of four TRIM genes in human macrophages.

Based on the GO analysis, the TRIM genes identified as up- or down-regulated by IFNs and/or IL27 could form a complex network that appears to be involved in a broad range of biological processes. These include regulation of antiviral responses such as the defense response to viruses, the viral life cycle, regulation of viral entry into the host cell, negative regulation of viral processes, and positive regulation of NF-κB TF. Moreover, based on our results, we hypothesize that TRIMs function by interacting with each other to modulate the innate immune response and control of viral infections. The hypothesis is supported by the antiviral activity of IFN-dependent responses and the multiple roles of IL27 in eliminating viral infections, as previously described by us and others [29,43,44,83].

In this study, we utilized the GeneMANIA and STRING databases to conduct a correlation analysis of TRIM genes and their adjacent genes or proteins. Our analysis revealed that *TRIM19* (*PML*), *21*, *22*, *25*, *31*, and *69* could interact with genes or proteins that play key roles in viral infections, responses to IFNs, and cytokine-mediated responses. Results suggest that TRIM genes modulated by IFNs and IL27 could regulate innate immune responses and inflammation. Notably, the innate response serves as the first line of defense to restrict viral infections, mainly through the activation of PRRs such as TLRs and RLRs. Given that earlier studies have demonstrated the antiviral activities of various TRIMs [84], we focused on TRIMs whose function primarily involved innate immune signaling pathways: TLRs, RIG-like receptors, and cyclic GMP-AMP synthase (cGAS)-stimulator of IFN genes (STING). Activation of these pathways leads to the production of IFNs and IL27 that promote the induction of ISG expression [25,85]. For instance, *TRIM25* is essential for RIG-I-mediated antiviral activity [86]. Moreover, *TRIM21*, *22* and *31* were upregulated by TLR3 and TLR4 ligands, whereas *TRIM59* was significantly down-regulated [76]. In this last study, 16 TRIM genes were significantly up-regulated by 8 out of 9 TLR ligands. However, TRIM proteins are also involved in regulating the expression of TLRs. For instance, TRIM21 negatively regulates TLR3, 4, 7, and 9 and RLR signaling by modulating the activities of IKKs and interferon regulatory factors [57,87]. *TRIM32* can negatively regulate TLR3 and TLR4 signaling by targeting TRIF to TAX1-binding protein 1 (TAXBP1)-dependent autophagic degradation [88]. *TRIM56*, which was induced in response to all three types of IFNs but not by IL27, promotes transcriptional induction of pro-inflammatory cytokines and type I IFN through regulation of TLR and STING signaling pathways [89,90].

One study showed that *TRIM21* acts as a defense against invading viruses [91]. It also inhibits HBV infection by interacting with its DNA polymerase through the SPRY motif [92]. In addition, *TRIM21* ubiquitinates the nucleocapsid of the porcine epidemic diarrhea virus and inhibits its proliferation [93]. However, *TRIM21* can also facilitate virus infection or immune escape for some viruses. For instance, the human papillomavirus E7 protein recruits the E3 ligase *TRIM21* to ubiquitinate and degrade IFNγ-inducible protein 16, inhibiting cell pyroptosis and self-escape from immune surveillance [94]. Similarly, *TRIM21* regulates virus-induced cell pyroptosis through polyubiquitination of ISG12a [95]. Likewise, *TRIM21* suppresses type I IFN responses by inhibiting the STING/IRF3 signal pathway, thus enhancing herpes simplex virus-1 infection in corneal epithelial cells [96].

*TRIM22* was synergistically induced at 24 h in A549 cells in response to IFN-I and IFN-II [97], and Barr et al., (2008) reported that *TRIM22* inhibits HIV-1 particle production [98]. In the same way, *TRIM22* suppresses ZIKV replication by targeting NS1 and NS3 proteins for proteasomal degradation [99]. Furthermore, *TRIM22* cooperates with MHC Class II transactivator to inhibit transcription initiation and elongation of HIV-1 RNA [100]. *TRIM22* also inhibits influenza A, hepatitis B, and C viruses [58,101,102]. Further, it was suggested that upregulation of *TRIM22* in COVID-19 patients might contribute to controlling the SARS-CoV-2 infection, resulting in a milder infection with fewer severe symptoms [103]. Wang et al. (2021) reported that *TRIM22* had an inhibitory effect on respiratory syncytial virus (RSV) infection, and downregulation of *TRIM22* moderately enhanced RSV replication in infected cells [93].

*TRIM25* has been reported to play a role in cell proliferation and the innate antiviral response against many RNA viruses. It targets different proteins, such as RIG-I, to enhance the production of IFN-I [86]. Furthermore, *TRIM25* inhibits HBV replication by amplifying IFN signaling [104,105]. More recently, an interaction was reported between *TRIM25*-HBx and the *TRIM25*-pregenomic RNA-RIG-I complex, resulting in control of HBV replication [106]. Additionally, *TRIM25* binds incoming and nuclear IAV viral ribonucleoprotein particles, inhibiting viral polymerase [107].

Both in vivo and in vitro studies have shown that *TRIM69* decreases DENV infectivity and contributes to the inhibitory effect of IFN-I [108]. In addition, *TRIM69*-depleted mice were significantly more susceptible to infection by DENV. Similarly, *TRIM56* overexpression and knockdown experiments show that replication of DENV, YFV [109], and ZIKV [110] is restricted. In each case, virus restriction involved both the RING domain and NHL-like repeats, and the restriction was linked to a decrease in viral RNA replication [110]. Expression of *TRIM14*, *19*, *21*, *25*, *26*, and *34* has been reported to be up-regulated by influenza virus infection in human epithelial cell lines in an IFN-I-dependent manner [63]. *TRIM56* has antiviral activity against positive-sense single-stranded RNA viruses, including flavivirus, coronavirus, and retrovirus 106, and negative-sense single-stranded RNA viruses, such as influenza A and B [111]. Consistent with our findings, up-regulation of TRIM gene expression in human macrophages was mainly observed in response to treatment with IFNγ and LPS under conditions that resulted in the induction of IFNβ [80]. This regulation was not observed in human macrophages treated with IL4, which does not induce INFβ secretion [63].

We and others have reported that IL-27 signaling leading to AVP production is IL27RA, STAT1, and IRF3 dependent but STAT2 independent. Here, we found that IL27 exhibits a protective function against MAYV infection in MDMs, albeit less strongly than IFNβ, in a concentration-dependent manner. In agreement with our findings, Kwock et al. (2020) reported a similar protective function against subcutaneous ZIKV infection in mice without type I IFN signaling. Likewise, ZIKV, CHIKV, and DENV activate IL27 signaling, eliciting a robust antiviral response in infected MDMs independent of IFNs. The study by Imamichi et al. (2023) reported that IL27 inhibits HIV-1 infection and induces HIV-1-resistant dendritic cells. Additionally, when MDMs are treated with IL27 or IFNβ and infected with MAYV, they express cellular restriction factors such as *TRIM19*, *21*, *22*, and *69*. In the case of *TRIM19*, *21*, *22*, and *69*, we found that IL27 was the most potent modulator of transcription, while IFNβ induces very low mRNA levels of those TRIMs. This suggests that these factors may inhibit MAYV replication, as observed. However, we cannot correlate the antiviral activity of IL27 with the potential effect of a specific TRIM since no knockdown or knockout of a specific TRIM was performed and we cannot test their function after stimulation with IL27. It has already been reported that many of these TRIMs are stimulated upon exposure to certain cytokines, such as IFN-I. To our knowledge, we are the first to report the induction of expression of these TRIMs in response to IL27 treatment within the context of a viral infection. We propose that this IL27 enhancement of TRIM expression can influence MDM permissibility toward arbovirus infections in human primary macrophages based on our findings.

## 5. Limitations of the Study

While our interpretations are supported by the transcriptomic analysis and RT-qPCR results, a significant limitation is that the protein levels of these TRIMs were not confirmed in this study. Additionally, as mentioned earlier, the methodology used, stimulus time, and cell type can strongly influence the observed results. However, it is important to highlight that we recently reported that IL27 not only induced a significant expression of ISGs/AVPs in macrophages but also in various immune cell populations, including MDDC, naïve CD4+, or CD8+ T cells, and non-immune cell populations such as HNEK and astrocytes [39]. Despite these limitations, we demonstrated in vitro that IL27 reduces the replication of the MAYV.

## 6. Conclusions

Our findings reveal that IL27, similar to IFNs, induces the expression of TRIM genes in human macrophages. We identified three common TRIM genes (*TRIM19*, *21*, and *22*) induced by IL27 and all types of human IFNs. Additionally, we performed the first report of transcriptional regulation of *TRIM19*, *21*, *22*, and *69* in response to IL27 treatment. These results indicated that, like IFNs, IL27 modulates the transcriptional expression of various TRIM-family members involved in the induction of innate immunity and an antiviral response. Furthermore, the functional analysis demonstrated that, like IFN, IL27 reduced MAYV replication in MDMs. Together, these findings suggest that, at a functional level, IFNs and IL27 share many similarities. Moreover, identifying distinct TRIM groups and their differential expression in response to IFNs and IL27 treatment provides insights into the regulatory mechanisms underlying the antiviral response in human macrophages. Further study will reveal the physiological relevance of the IL27-regulated TRIM transcription and the involvement of other regulatory factors in the signaling pathway.

## Figures and Tables

**Figure 1 viruses-16-00996-f001:**
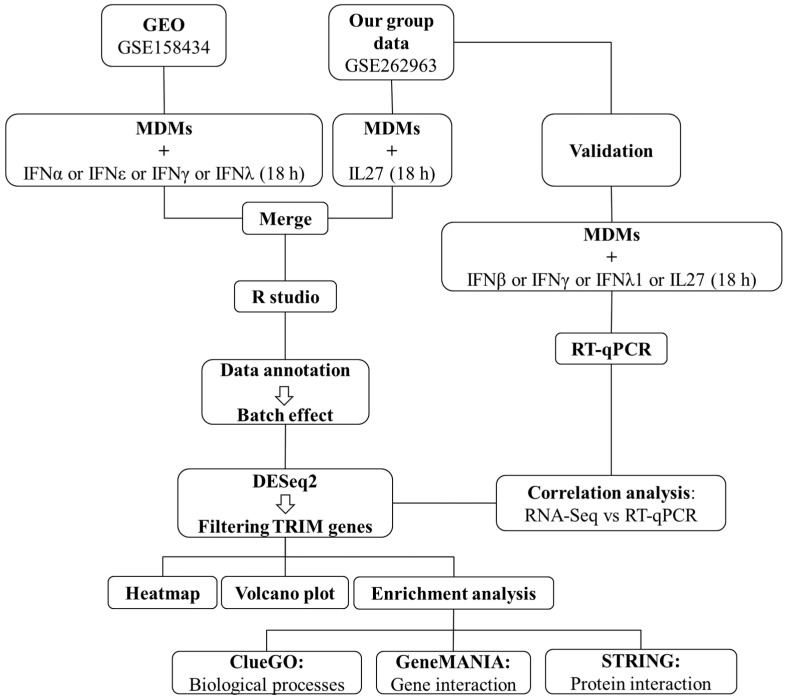
Study workflow.

**Figure 2 viruses-16-00996-f002:**
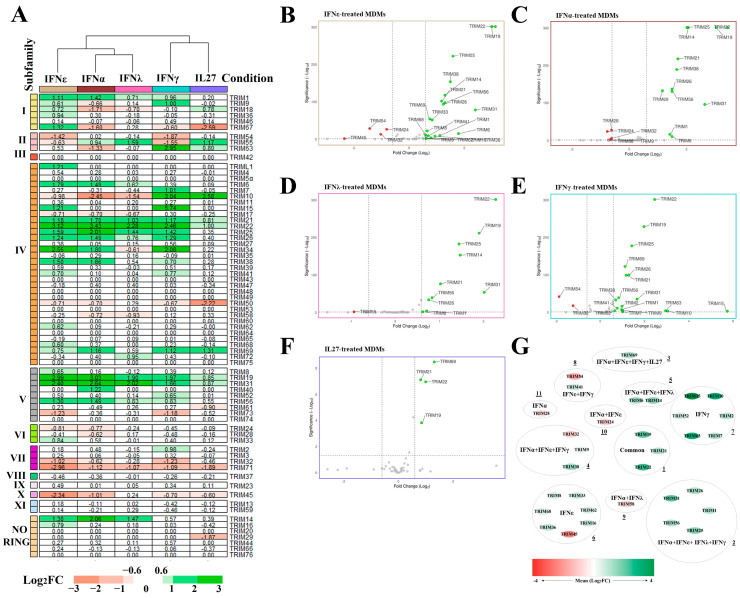
TRIMs expression of MDMs treated with IFNα, IFNε, IFNγ, IFNλ, and IL27. The following RNA-seqs were reanalyzed: GSE262963 (GEO)) for MDMs treated or not with IL27; GSE158434 (GEO) for MDMs treated or not with IFNα, IFNε, IFNγ, and IFNλ. (**A**) Heat map of TRIM expression in MDMs treated with IFNα, IFNε, IFNγ, IFNλ, and IL27. The heat map was obtained from R software (version 4.2.0) [64], where each column represents a condition (treatment to MDM) with a color: IFNα (brown), IFNγ (turquoise), IFNλ (pink), and IL27 (blue-violet). The heat map shows rows with Log_2_FC values for 72 selected TRIMs (including subfamilies) indicating up-regulation (green) or down-regulation (red) of their expression. MA plots of differentially expressed genes (DEGs) of TRIMs. Each MA plot shows DEGs of TRIMs in IFNε-treated MDMs (**B**), DEGs of TRIMs in IFNα-treated MDMs (**C**), DEGs of TRIMs in IFNλ-treated MDMs (**D**), DEGs of TRIMs in IFNγ-treated MDMs (**E**), and DEGs of TRIMs in IL27-treated MDMs (**F**). (**G**) Gene-gene association network DEGs of TRIM expression in MDMs treated or not with IFNα, IFNε, IFNγ, IFNλ, and IL27. The network was obtained from Cytoscape. Genes are shown as an ellipse, grouped according to the treatment(s) in MDMs. The node colors represent the up-regulation (green) or down-regulation (red) of TRIM expression based on the mean (Log_2_FC).

**Figure 3 viruses-16-00996-f003:**
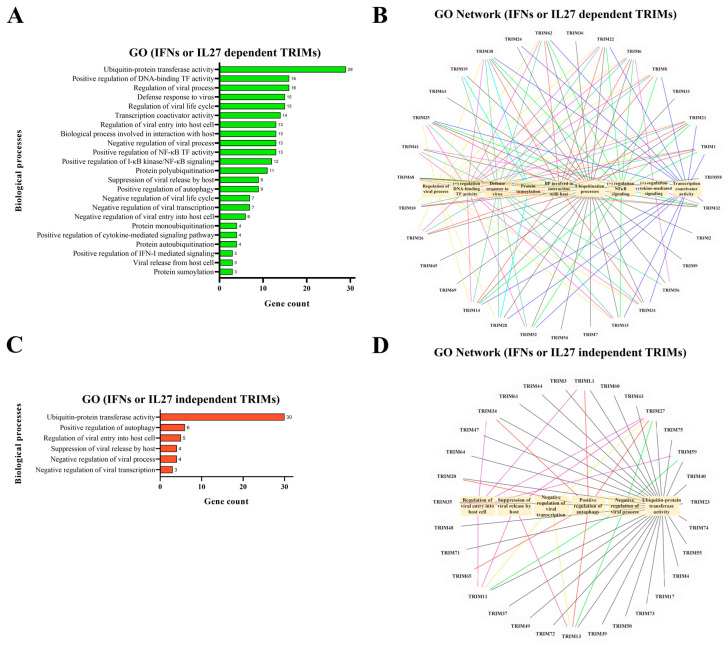
DEGs expression of IFNs or IL27-dependent or independent TRIM in MDMs. The following RNA-seqs were reanalyzed: GSE262963 (GEO) for MDMs treated or not with IL27; GSE158434 (GEO) for MDMs treated or not with IFNα, IFNε, IFNγ, IFNλ. Gene Ontology (GO) analysis was achieved in Cytoscape using the plugin ClueGO. Each GO diagram shows the common biological processes implicated in IFNs or IL27-dependent TRIMs (**A**) and the common biological processes implicated in IFNs or IL27-independent TRIMs (**B**). The number of genes regulated (the gene count) is shown for each biological process. Each GO network was realized using Cytoscape, which displays each TRIM involved in IFN or IL27-dependent biological processes (**C**) and the TRIM involved in IFN or IL27-independent biological processes (**D**). Each biological process is represented by an edge of a specific color.

**Figure 4 viruses-16-00996-f004:**
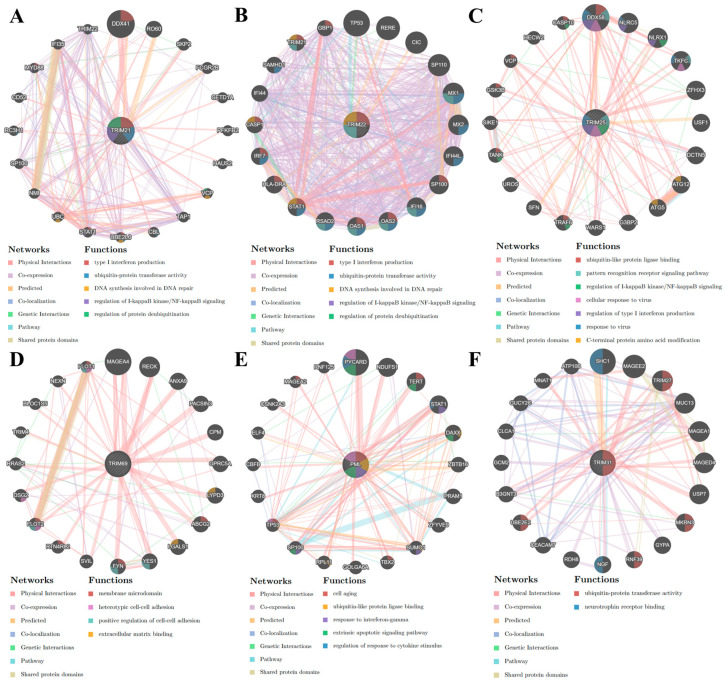
The gene-gene interaction network of TRIM significantly up-regulated the transcripts of MDMs from the transcriptomic analysis. Each network of significantly up-regulated TRIM genes, such as *TRIM21* (**A**), *TRIM22* (**B**), *TRIM25* (**C**), *TRIM69* (**D**), *TRIM19/PML* (**E**), and *TRIM31* (**F**), was obtained from GeneMANIA. The functions of the genes are represented by colored sections in the circle nodes, with the corresponding functions indicated in the legend labeled “Functions” using a color code. Meanwhile, the types of interactions between genes are represented by the color of the lines, and the legend labeled “Networks” provides a color code for understanding the different types of interactions. Additionally, the size of each node corresponds to the strength of the interaction.

**Figure 5 viruses-16-00996-f005:**
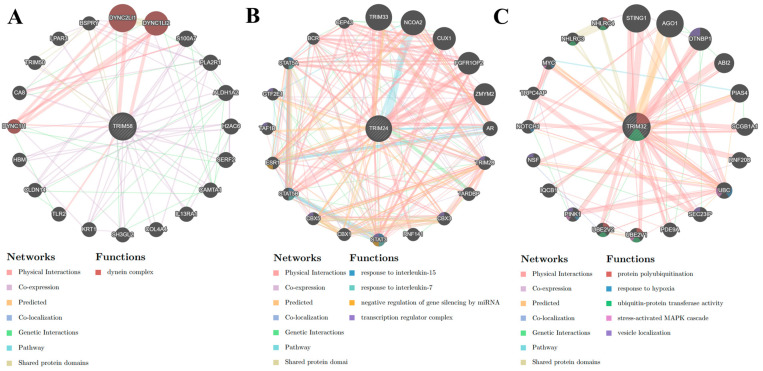
The gene-gene interaction network of TRIM significantly down-regulated the transcripts of MDMs from the transcriptomic analysis. Each network of significantly down-regulated TRIM genes, such as *TRIM58* (**A**), *TRIM24* (**B**), and *TRIM32* (**C**), was obtained from GeneMANIA. The functions of the genes are represented by colored sections in the circle nodes, with the corresponding functions indicated in the legend labeled “Functions” using a color code. Meanwhile, the types of interactions between genes are represented by the color of the lines, and the legend labeled “Networks” provides a color code for understanding the different types of interactions. Additionally, the size of each node corresponds to the strength of the interaction.

**Figure 6 viruses-16-00996-f006:**
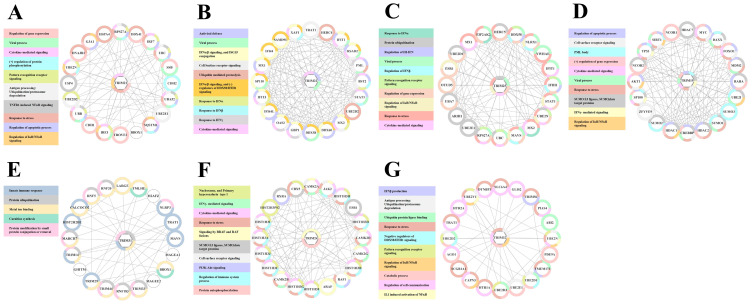
Protein-protein interaction network of TRIM-regulated transcripts of MDMs from the transcriptomic analysis. The STRING output shows 20 interactors for each analysis based on the selected TRIM and its confidence threshold. TRIM21 with 0.7 of confidence (**A**), TRIM22 with 0.4 of confidence (**B**), TRIM25 with 0.7 of confidence (**C**), TRIM19 with 0.7 of confidence (**D**), *TRIM31* with 0.7 of confidence (**E**), TRIM24 with 0.7 of confidence (**F**), and TRIM32 with 0.7 of confidence (**G**). Nodes are proteins; the edges represent protein-protein interactions contributing to a shared function. The full donut represents the functional enrichment analysis.

**Figure 7 viruses-16-00996-f007:**
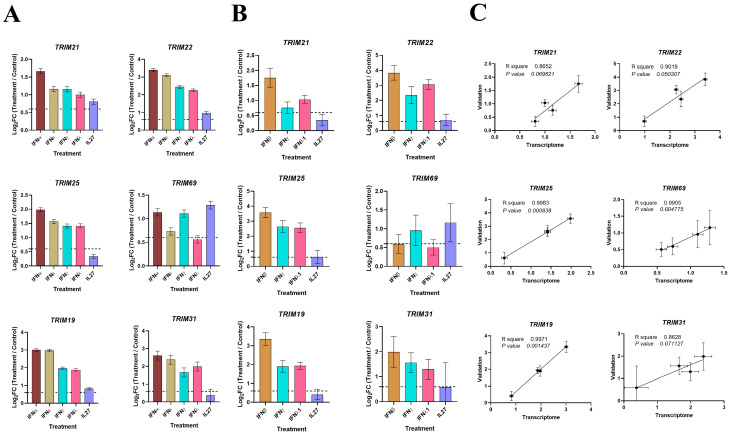
Expression of common TRIMs among treatments with IFNs or IL27 in MDMs. (**A**) TRIM expression in MDMs treated or not with IFNα, IFNε, IFNγ, IFNλ, and IL27. The following RNA-seq were reanalyzed: GSE262963 (GEO) for MDMs treated or not with IL27; GSE158434 (GEO) for MDMs treated or not with IFNα, IFNε, IFNγ, and IFNλ. Logarithm of fold change (Log_2_FC) ratios for *TRIM21*, *TRIM22*, *TRIM25*, *TRIM69*, *TRIM19*, and *TRIM31* in Treatment/Control. n = 3. Data are represented as mean ± SEM. (**B**) TRIM expression in MDMs treated or not with IFNβ, IFNγ, IFNλ1, and IL27. Cell lysates were obtained at 24 h post-treatment, and RT-qPCR was performed. Logarithm of fold change (Log_2_FC) ratios for *TRIM21*, *TRIM22*, *TRIM25*, *TRIM69*, *TRIM19*, and *TRIM31* in Treatment/Control. n = 4, without technical triplicates. Data are represented as mean ± SEM. A Log_2_FC of 0.6 and −0.6 were considered as up-regulation or down-regulation of gene expression, respectively (dotted lines). Linear regression analysis of Log_2_FC ratios between mRNA level transcriptomes and mRNA level validation was performed for each TRIM (**C**). The R square is shown.

**Figure 8 viruses-16-00996-f008:**
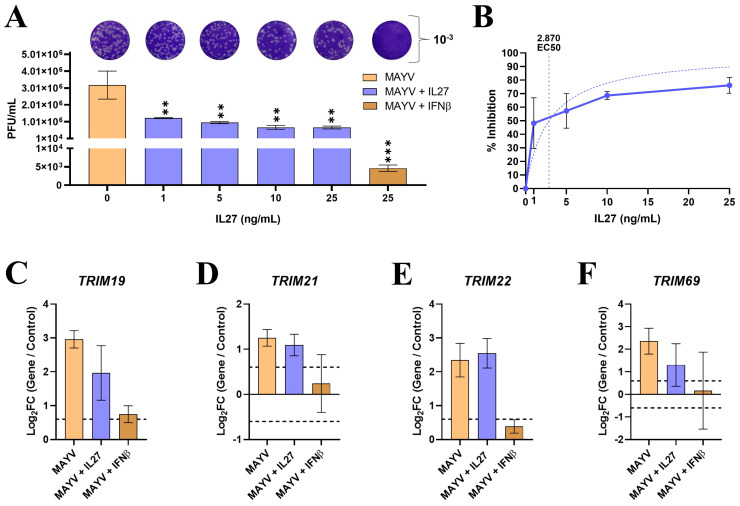
MAYV replication in human macrophages pre-treated with IL27 and expression of *TRIM19*, *21*, *22*, and *69* and infected with MAYV. Human macrophages were pre-treated with IL27 (1, 5, 10, 25 ng/mL) or IFNβ (25 ng/mL) for 6 h, followed by infection with MAYV at a MOI of 0.5. Culture supernatants and cells were collected at 24 h.p.i. Viral titration was determined by plaque assay on Vero cells using culture supernatants, while cell lysates underwent RT-qPCR analysis. MAYV replication in MDMs treated with IL27 (**A**). Percentage inhibition of MAYV replication and EC50 of IL27 in MDMs (**B**). The logarithm of fold change (Log_2_FC) ratios for *TRIM19* (**C**), *TRIM21* (**D**), *TRIM22* (**E**), and *TRIM69* (**F**) relative to the control are shown. The study included three replicates (n = 3, without technical triplicates), and the data are presented as mean ± SEM. A Log_2_FC of 0.6 and −0.6 indicated up-regulation and down-regulation of gene expression, respectively (dotted lines). Asterisks refer to statistically significant differences between control and treatments as: ** *p* < 0.01, *** *p* < 0.001.

**Table 1 viruses-16-00996-t001:** Summary of TRIM expression in macrophages infected with viruses or stimulated with IFNs or IL27.

	**Tripartite Motif (TRIM) Family Members Expression in Macrophages**									
	** *Sp.* **	***Homo sapiens* (Human)**	***Mus musculus* (C57BL/6mice)**	***Homo sapiens* (Human)**	***Homo sapiens* (Human)**	***Homo sapiens* (Human)**	***Homo sapiens* (Human)**	***Homo sapiens* (Human)**	***Homo sapiens* (Human)**	***Homo sapiens* (Human)**
	**Cell type**	MDMs FCS + M-CSF, 7 d.d.	BMDM M-CSF	MDMs h-AB serum, 7 to 11 d.d.	MDMs h-AB serum, 7 to 11 d.d.	MDMs h-AB serum, 7 d.d.	MDMs h-AB serum, 7 d.d.	MDMs h-AB serum, 7 d.d.	MDMs h-AB serum, 7 d.d.	MDMs FBS, 7 d.d.
	**Stimulus**	100 ng/ml LPS + 20 ng/ml IFNγ 18 h	Influenza virus (IFNβ-dependent antiviral response), 24 hpi.	1000 UI/ml universal IFN-I (recombinant IFNαA-E), 8 h	1000 UI/ml IFN-II (IFNγ), 8 h	25 gr/mL IFN-I (IFNα), 18 h	25 ng/mL IFN-I (IFNε), 18 h	25 ng/mL IFN-II (IFNγ), 18 h	25 ng/mL IFN-III (IFNλ), 18 h	25 ng/mL Interleukin 27, 18 h
	**Technique**	Microarray	RT-qPCR	RT-PCR Array	RT-PCR Array	Bulk RNA-Seq. and/or RT-qPCR	Bulk RNA-Seq. and/or RT-qPCR	Bulk RNA-Seq. and/or RT-qPCR	Bulk RNA-Seq. and/or RT-qPCR	Bulk RNA-Seq. and/or RT-qPCR
	**Reference.**	[77]	[78]	[73]	[73]	Our study	Our study	Our study	Our study	Our study
**Sub** **Family**	**TRIM Gene**	**mRNA Expression in Macrophages**								
**I**	**TRIM1**	**N.C.**	**N.C.**	**N.C.**	**N.C.**	** Up-regulated **	** Up-regulated **	** Up-regulated **	** Up-regulated **	**N.C.**
	**TRIM9**	**N.R.**	**N.C.**	**N.C.**	** Down-regulated **	** Down-regulated **	** Up-regulated **	** Up-regulated **	**N.C.**	**N.C.**
	**TRIM18**	** Up-regulated **	** Up-regulated **	**N.D.**	**N.D.**	No significant, Down-regulated	No significant, Up-regulated	**N.C.**	No significant, Down-regulated	No significant, Up-regulated
	**TRIM36**	**N.R**	**N.D.**	**N.D.**	**N.D.**	**N.C.**	** Up-regulated **	**N.C.**	**N.C.**	**N.C.**
	**TRIM46**	** Up-regulated **	** Up-regulated **	**N.D.**	**N.D.**	**N.C.**	**N.C.**	**N.C.**	**N.C.**	**N.C.**
	**TRIM67**	**N.R.**	**N.R.**	**N.D.**	**N.D.**	No significant, Down-regulated	No significant, Up-regulated	No significant, Down-regulated	**N.C.**	No significant, Down-regulated
**II**	**TRIM54**	**N.R.**	**N.R.**	** Down-regulated **	** Down-regulated **	**N.C.**	** Down-regulated **	** Down-regulated **	**N.C.**	**N.C.**
	**TRIM55**	**N.R.**	**N.R.**	**N.D.**	**N.D.**	No significant, Up-regulated	No significant, Down-regulated	No significant, Down-regulated	No significant, Up-regulated	No significant, Up-regulated
	**TRIM63**	**N.R.**	**N.R.**	**N.D.**	**N.D.**	No significant, Down-regulated	**N.C.**	No significant, Up-regulated	**N.C.**	No significant, Up-regulated
**III**	**TRIM42**	**N.R.**	**N.R.**	**N.D.**	**N.D.**	**N.C.**	**N.C.**	**N.C.**	**N.C.**	**N.C.**
**IV**	**TRIML1**	**N.R.**	**N.R.**	**N.D.**	**N.D.**	**N.C.**	No significant, Up-regulated	**N.C.**	**N.C.**	**N.C.**
	**TRIM4**	**N.R.**	**N.R.**	**N.C.**	** Down-regulated **	**N.C.**	**N.C.**	**N.C.**	**N.C.**	**N.C.**
	**TRIM5** **α**	** Up-regulated **	**No homologue in mouse**	** Up-regulated **	**N.C.**	**N.C.**	**N.C.**	**N.C.**	**N.C.**	**N.C.**
	**TRIM6**	** Up-regulated **	** Up-regulated **	**N.D.**	**N.D.**	** Up-regulated **	** Up-regulated **	**N.C.**	** Up-regulated **	**N.C.**
	**TRIM7**	**N.R**	**N.R.**	**N.D.**	**N.D.**	**N.C.**	**N.C.**	** Up-regulated **	**N.C.**	**N.C.**
	**TRIM10**	** Up-regulated **	**N.D.**	**N.D.**	**N.D.**	No significant, Down-regulated	No significant, Down-regulated	** Up-regulated **	No significant, Down-regulated	No significant, Up-regulated
	**TRIM11**	**N.R.**	**N.R.**	**N.C.**	**N.C.**	**N.C.**	**N.C.**	**N.C.**	**N.C.**	**N.C.**
	**TRIM15**	**N.C.**	**N.D.**	**N.D.**	**N.D.**	**N.C.**	No significant, Up-regulated	** Up-regulated **	**N.C.**	**N.C.**
	**TRIM17**	** Up-regulated **	**N.D.**	**N.D.**	**N.D.**	No significant, Down-regulated	No significant, Down-regulated	**N.C.**	No significant, Down-regulated	**N.C.**
	**TRIM21**	** Up-regulated **	** Up-regulated **	** Up-regulated **	** Up-regulated **	** Up-regulated **	** Up-regulated **	** Up-regulated **	** Up-regulated **	** Up-regulated **
	**TRIM22**	** Up-regulated **	**No homologue in mouse**	** Up-regulated **	** Up-regulated **	** Up-regulated **	** Up-regulated **	** Up-regulated **	** Up-regulated **	** Up-regulated **
	**TRIM25**	** Up-regulated **	** Up-regulated **	** Up-regulated **	** Up-regulated **	** Up-regulated **	** Up-regulated **	** Up-regulated **	** Up-regulated **	**N.C.**
	**TRIM26**	** Up-regulated **	** Up-regulated **	** Up-regulated **	**N.C.**	** Up-regulated **	** Up-regulated **	** Up-regulated **	** Up-regulated **	**N.C.**
	**TRIM27**	** Down-regulated **	**N.C.**	**N.D.**	**N.D.**	**N.C.**	**N.C.**	**N.C.**	**N.C.**	**N.C.**
	**TRIM34**	** Up-regulated **	** Up-regulated **	** Up-regulated **	**N.C.**	No significant, Up-regulated	No significant, Up-regulated	No significant, Up-regulated	No significant, Down-regulated	**N.C.**
	**TRIM35**	** Up-regulated **	** Up-regulated **	** Up-regulated **	**N.D.**	**N.C.**	**N.C.**	**N.C.**	**N.C.**	**N.C.**
	**TRIM38**	** Up-regulated **	**N.C.**	** Up-regulated **	**N.C.**	** Up-regulated **	** Up-regulated **	** Up-regulated **	**N.C.**	**N.C.**
	**TRIM39**	**N.C.**	**N.C.**	**N.C.**	**N.C.**	**N.C.**	** Up-regulated **	**N.C.**	**N.C.**	**N.C.**
	**TRIM41**	**N.R.**	**N.R.**	**N.D.**	**N.D.**	**N.C.**	No significant, Up-regulated	** Up-regulated **	**N.C.**	**N.C.**
	**TRIM43**	**N.R.**	**N.R.**	**N.D.**	**N.D.**	**N.C.**	**N.C.**	**N.C.**	**N.C.**	**N.C.**
	**TRIM47**	**N.R.**	**N.R.**	**N.C.**	**N.C.**	**N.C.**	**N.C.**	**N.C.**	**N.C.**	**N.C.**
	**TRIM48**	** Up-regulated **	**No homologue in mouse**	**N.D.**	**N.D.**	**N.C.**	**N.C.**	**N.C.**	**N.C.**	**N.C.**
	**TRIM49**	**N.C.**	**No homologue in mouse**	**N.D.**	**N.D.**	**N.C.**	**N.C.**	**N.C.**	**N.C.**	**N.C.**
	**TRIM50**	**N.R.**	**N.R.**	**N.D.**	**N.D.**	No significant, Down-regulated	No significant, Down-regulated	No significant, Down-regulated	**N.C.**	No significant, Down-regulated
	**TRIM53**	**N.R.**	**N.R.**	**N.D.**	**N.D.**	**N.C.**	**N.C.**	**N.C.**	**N.C.**	**N.C.**
	**TRIM58**	**N.C.**	**N.D.**	** Up-regulated **	**N.C.**	** Down-regulated **	**N.C.**	**N.C.**	** Down-regulated **	**N.C.**
	**TRIM60**	**N.R.**	**N.R.**	**N.D.**	**N.D.**	**N.C.**	**N.C.**	**N.C.**	**N.C.**	**N.C.**
	**TRIM62**	** Up-regulated **	**N.D.**	**N.C.**	**N.C.**	**N.C.**	** Up-regulated **	**N.C.**	**N.C.**	**N.C.**
	**TRIM64**	**N.R.**	**N.R.**	**N.D.**	**N.D.**	**N.C.**	**N.C.**	**N.C.**	**N.C.**	**N.C.**
	**TRIM65**	**N.R.**	**N.R.**	**N.C.**	**N.C.**	**N.C.**	**N.C.**	**N.C.**	**N.C.**	**N.C.**
	**TRIM68**	** Up-regulated **	**N.C.**	**N.C.**	**N.C.**	**N.C.**	** Up-regulated **	**N.C.**	**N.C.**	**N.C.**
	**TRIM69**	**N.R.**	**N.R.**	** Up-regulated **	** Up-regulated **	** Up-regulated **	** Up-regulated **	** Up-regulated **	**N.C.**	** Up-regulated **
	**TRIM72**	**N.R.**	**N.R.**	**N.D.**	**N.D.**	**N.C.**	**N.C.**	**N.C.**	No significant, Up-regulated	**N.C.**
	**TRIM75**	**N.R.**	**N.R.**	**N.D.**	**N.D.**	**N.C.**	**N.C.**	**N.C.**	**N.C.**	**N.C.**
**V**	**TRIM8**	** Up-regulated **	** Up-regulated **	**N.C.**	**N.C.**	**N.C.**	** Up-regulated **	**N.C.**	**N.C.**	**N.C.**
	**TRIM19**	** Up-regulated **	** Up-regulated **	** Up-regulated **	** Up-regulated **	** Up-regulated **	** Up-regulated **	** Up-regulated **	** Up-regulated **	** Up-regulated **
	**TRIM31**	** Up-regulated **	**N.D.**	** Up-regulated **	**N.D.**	** Up-regulated **	** Up-regulated **	** Up-regulated **	** Up-regulated **	No significant, Up-regulated
	**TRIM40**	**N.R.**	**N.R.**	**N.D.**	**N.D.**	No significant, Up-regulated	**N.C.**	**N.C.**	**N.C.**	**N.C.**
	**TRIM52**	** Down-regulated **	**N.D.**	**N.C.**	**N.C.**	**N.C.**	**N.C.**	** Up-regulated **	**N.C.**	**N.C.**
	**TRIM56**	**N.R.**	**N.R.**	** Up-regulated **	** Up-regulated **	** Up-regulated **	** Up-regulated **	No significant, Up-regulated	** Up-regulated **	**N.C.**
	**TRIM61**	**N.R.**	**N.R.**	**N.D.**	**N.D.**	**N.C.**	**N.C.**	**N.C.**	**N.C.**	No significant, Down-regulated
	**TRIM73**	**N.R.**	**N.R.**	**N.C.**	**N.C.**	**N.C.**	No significant, Down-regulated	No significant, Down-regulated	**N.C.**	**N.C.**
	**TRIM74**	**N.R.**	**N.R.**	**N.D.**	**N.D.**	**N.C.**	**N.C.**	**N.C.**	**N.C.**	**N.C.**
**VI**	**TRIM24**	**N.C.**	**N.C.**	**N.C.**	**N.C.**	** Down-regulated **	** Down-regulated **	**N.C.**	**N.C.**	**N.C.**
	**TRIM28**	**N.C.**	**N.C.**	** Down-regulated **	** Down-regulated **	** Down-regulated **	**N.C.**	**N.C.**	**N.C.**	**N.C.**
	**TRIM33**	** Up-regulated **	**N.D.**	**N.C.**	**N.C.**	**N.C.**	** Up-regulated **	**N.C.**	**N.C.**	**N.C.**
**VII**	**TRIM2**	**N.R.**	**N.R.**	**N.C.**	** Down-regulated **	**N.C.**	**N.C.**	No significant, Up-regulated	**N.C.**	**N.C.**
	**TRIM3**	** Up-regulated **	** Up-regulated **	**N.C.**	**N.C.**	**N.C.**	**N.C.**	**N.C.**	**N.C.**	**N.C.**
	**TRIM32**	** Down-regulated **	**N.D.**	**N.C.**	** Down-regulated **	** Down-regulated **	** Down-regulated **	** Down-regulated **	**N.C.**	**N.C.**
	**TRIM71**	**N.R.**	**N.R.**	**N.D.**	**N.D.**	No significant, Down-regulated	No significant, Down-regulated	No significant, Down-regulated	No significant, Up-regulated	No significant, Down-regulated
**VIII**	**TRIM37**	**N.C.**	**N.C.**	** Down-regulated **	** Down-regulated **	**N.C.**	**N.C.**	**N.C.**	**N.C.**	**N.C.**
**IX**	**TRIM23**	** Down-regulated **	** Up-regulated **	**N.D.**	**N.D.**	**N.C.**	**N.C.**	**N.C.**	**N.C.**	**N.C.**
**X**	**TRIM45**	** Up-regulated **	** Up-regulated **	**N.D.**	**N.D.**	No significant, Down-regulated	** Down-regulated **	No significant, Down-regulated	**N.C.**	No significant, Down-regulated
**XI**	**TRIM13**	** Up-regulated **	**N.D.**	**N.D.**	**N.C.**	**N.C.**	**N.C.**	**N.C.**	**N.C.**	**N.C.**
	**TRIM59**	**N.C.**	**N.C.**	** Down-regulated **	** Down-regulated **	**N.C.**	**N.C.**	**N.C.**	**N.C.**	**N.C.**
**NO RING**	**TRIM14**	** Up-regulated **	** Up-regulated **	** Up-regulated **	**N.C.**	** Up-regulated **	** Up-regulated **	**N.C.**	** Up-regulated **	**N.C.**
	**TRIM16**	** Down-regulated **	** Down-regulated **	**N.C.**	** Down-regulated **	**N.C.**	** Up-regulated **	**N.C.**	**N.C.**	**N.C.**
	**TRIM20**	**N.C.**	** Up-regulated **	** Up-regulated **	** Up-regulated **	**N.C.**	**N.C.**	**N.C.**	**N.C.**	**N.C.**
	**TRIM29**	** Up-regulated **	**N.D.**	**N.D.**	**N.D.**	**N.C.**	**N.C.**	**N.C.**	**N.C.**	No significant, Down-regulated
	**TRIM44**	** Down-regulated **	**N.C.**	**N.C.**	**N.C.**	**N.C.**	**N.C.**	**N.C.**	**N.C.**	**N.C.**
	**TRIM66**	** Down-regulated **	**N.D.**	** Down-regulated **	** Down-regulated **	**N.C.**	**N.C.**	**N.C.**	**N.C.**	**N.C.**
	**TRIM76**	**N.R.**	**N.R.**	**N.D.**	**N.D.**	**N.C.**	**N.C.**	**N.C.**	**N.C.**	**N.C.**

**N.R.** = No reported/ no evaluated. **N.D.** = Evaluated but was not detected and/or expressed in macrophages. **N.C.** = Expressed in macrophages, but without change in gene expression in response to stimuli. d.d. = days of differentiation.

## Data Availability

We utilized two datasets: the first was obtained from the Gene Expression Omnibus (GEO) database under accession number GSE158434, while the second dataset was derived from our own research with accession number GSE262963.

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
