# Peer review of "Interleukin 27, Similar to Interferons, Modulates Gene Expression of Tripartite Motif (TRIM) Family Members and Interferes with Mayaro Virus Replication in Human Macrophages"

_viruses, 2024, doi:10.3390/v16060996_

Round 1

Reviewer 1 Report (Previous Reviewer 1)

Comments and Suggestions for Authors

After 2 rounds of revision, the authors finally consented to perform an additional experiment, in order to consolidate their manuscript. They therefore assessed the effect of IL27 on Mayaro virus replication.  They also evaluated the effect of infection +/- IL27 or IFN on the transcription of some TRIM genes.

Unfortunately, these new results do not add much to the study. In particular, they do not allow the identification of which TRIM proteins are involved in the antiviral activity of IL27, which has been my main request from the beginning.

Minor comment

Lines 366-367: "Among them, we included TRIM19, TRIM21, and TRIM22, which were induced by Il27 as well as all three types of human IFNs; TRIM19 and 25 which were induced only by IFNs, but not by IL27" 

The information is contradictory. Is TRIM19 induced by IL27 and IFN or just by IL27?

Author Response

Review 1 Comments

After 2 rounds of revision, the authors finally consented to perform an additional experiment, in order to consolidate their manuscript. They therefore assessed the effect of IL27 on Mayaro virus replication.  They also evaluated the effect of infection +/- IL27 or IFN on the transcription of some TRIM genes.

Unfortunately, these new results do not add much to the study. In particular, they do not allow the identification of which TRIM proteins are involved in the antiviral activity of IL27, which has been my main request from the beginning.

Answer: We agree with the reviewer. In the present study, we did not evaluate the expression of the TRIMs of interest at the protein level since our initial study aimed to evaluate the effect of IL27 treatment on the transcriptional expression of TRIMs. However, in a future study, we are confident that we will be able to evaluate this effect at the protein level and determine the antiviral activity of the TRIMs induced by IL27. The TRIMs of interest will be silenced to achieve this, and their effect on replicating the viruses of interest will be evaluated.

Minor comment

Lines 366-367: "Among them, we included TRIM19, TRIM21, and TRIM22, which were induced by Il27 as well as all three types of human IFNs; TRIM19 and 25 which were induced only by IFNs, but not by IL27" 

The information is contradictory. Is TRIM19 induced by IL27 and IFN or just by IL27?

Answer: We apologize for any confusion, and we want to assure the reviewer that we have made the necessary corrections in Lines 354 and 355 to ensure the accuracy of our findings.

Reviewer 2 Report (Previous Reviewer 2)

Comments and Suggestions for Authors

In this study, the authors aim to address the gene regulation of TRIM family induced by Type-I,II,III interferons and IL-27 via bulk RNA sequencing approach. They highlight that the induction of TRIM genes is caused in monocyte-derived macrophages following treatment of interferons and IL-27, as evidenced by computational analysis. Additionally, they quantify the TRIM gene expression using qRT-PCR. Although overall the manuscript is well written, there are some minor concerns to reach the standard of Viruses.

- Minor revision

* In Fig. 8, the expression levels of TRIM 19, TRIM 21, TRIM 22 and TRIM 69 should be required for macrophages treated IL27 and Interferon-beta alone for measuring the effect of MAYV on TRIM expression in IL27 and interferon-beta treated macrophages.

* There are man typographical errors that need to be corrected.

Comments on the Quality of English Language

The quality of English language is excellent.

Author Response

Review 2

Comments

In this study, the authors aim to address the gene regulation of TRIM family induced by Type-I,II,III interferons and IL-27 via bulk RNA sequencing approach. They highlight that the induction of TRIM genes is caused in monocyte-derived macrophages following treatment of interferons and IL-27, as evidenced by computational analysis. Additionally, they quantify the TRIM gene expression using qRT-PCR. Although overall the manuscript is well written, there are some minor concerns to reach the standard of Viruses.

- Minor revision

* In Fig. 8, the expression levels of TRIM 19, TRIM 21, TRIM 22 and TRIM 69 should be required for macrophages treated IL27 and Interferon-beta alone for measuring the effect of MAYV on TRIM expression in IL27 and interferon-beta treated macrophages.

Answer: Given that the analysis of the transcriptomes of macrophages treated with IFNs and IL27 and validation by RT-qPCR using cells treated with IFNs or IL27 without infection shows an increase in TRIM mRNA levels, our next aim was to evaluate whether both IL27 and IFNs decreased the replication of the Mayaro virus (Fig. 8). Therefore, given that we already had the results of the effect of IFNs and IL27, without infection on the expression of TRIMs (Fig. 7), we consider that it is not necessary to include those controls for Figure 8.

* There are man typographical errors that need to be corrected.

Answer: Thank you for your observation. Ajit Kumar from George Washington University, D.C., reviewed the manuscript for typographical errors.

This manuscript is a resubmission of an earlier submission. The following is a list of the peer review reports and author responses from that submission.

Round 1

Reviewer 1 Report

Comments and Suggestions for Authors

In this article, Hernandez-Sarmiento and colleagues evaluated the induction of TRIM gene expression by type I, II and III interferons and by IL27 in human macrophages. They first analyzed TRIM genes that are regulated by IFNs or IL27 using bulk RNAseq. They then used several algorithms to illustrate the identity of the genes whose expression was found to be regulated. Finally, they confirmed the results obtained by RT-qPCR experiments performed on 6 of the TRIMs transcripts. A table summarizes the results obtained and compares them to previous studies.

General comments

This article unfortunately does not contribute much to the discipline. The RNAseq data only confirms data from 20 years ago on the induction of TRIM genes by interferon. Importantly, these data were not generated by the authors, but were extracted from the GEO database. These RNAseq data were in fact obtained by Szaniawski M and Planelles V (Huntsman Cancer Instituten, Salt Lake City, USA). The only original data in this paper is therefore the expression of TRIM genes following IL27 treatment. However, the identification of 4 TRIM genes induced by IL27 in macrophages was not followed by any functional experiment. IL27 is known to induce an antiviral response and it would have been interesting to determine whether any of the 4 TRIM proteins identified could play a role in this activity.

Figures 2, 3, 4 and 5 are of no use, since the authors only used free software found on the internet, such as GenMANIA or STRING, to obtain information on the identified TRIM genes. These TRIM genes are however very well characterized and a bibliographic analysis would have been sufficient.

In summary, the article attempts to hide the lack of results with an exaggerated and somewhat naive use of online algorithms.

Specific comments

Line 133: I don't think those are human macrophages in the article Rajsbaum et al. 2008. Data on human macrophages are from Martinez FO et al. J. Immunol. 2006

Line 202: Doses of interferon should be indicated in IU/ml. Concentrations in ng/ml mean nothing because it depends on the specific activity of interferons.

Line 203: Which IFN-l ? Where are the references for IFN-a and -w ?

Figure 1A should include some positive and negative controls. But, once again, the data presented were not generated by the authors, with the exception of IL27.

Table 1 is difficult to read and contains typos: "ngr/mL" instead of ng/mL, "Influeza" instead of "Influenza"

Comments on the Quality of English Language

The quality of the English language is generally correct.

Author Response

Reviewer 1

Open Review

Open Review

Quality of English Language

( ) I am not qualified to assess the quality of English in this paper
( ) English very difficult to understand/incomprehensible
( ) Extensive editing of English language required
(x) Moderate editing of English language required
( ) Minor editing of English language required
( ) English language fine. No issues detected

Yes

Can be improved

Must be improved

Not applicable

Does the introduction provide sufficient background and include all relevant references?

(x)

( )

( )

( )

Are all the cited references relevant to the research?

(x)

( )

( )

( )

Is the research design appropriate?

( )

( )

(x)

( )

Are the methods adequately described?

( )

( )

(x)

( )

Are the results clearly presented?

( )

( )

(x)

( )

Are the conclusions supported by the results?

( )

( )

(x)

( )

Comments and Suggestions for Authors

In this article, Hernandez-Sarmiento and colleagues evaluated the induction of TRIM gene expression by type I, II and III interferons and by IL27 in human macrophages. They first analyzed TRIM genes that are regulated by IFNs or IL27 using bulk RNAseq. They then used several algorithms to illustrate the identity of the genes whose expression was found to be regulated. Finally, they confirmed the results obtained by RT-qPCR experiments performed on 6 of the TRIMs transcripts. A table summarizes the results obtained and compares them to previous studies.

General comments

This article unfortunately does not contribute much to the discipline. The RNAseq data only confirms data from 20 years ago on the induction of TRIM genes by interferon. Importantly, these data were not generated by the authors, but were extracted from the GEO database. These RNAseq data were in fact obtained by Szaniawski M and Planelles V (Huntsman Cancer Instituten, Salt Lake City, USA). The only original data in this paper is therefore the expression of TRIM genes following IL27 treatment. However, the identification of 4 TRIM genes induced by IL27 in macrophages was not followed by any functional experiment. IL27 is known to induce an antiviral response and it would have been interesting to determine whether any of the 4 TRIM proteins identified could play a role in this activity.

Figures 2, 3, 4 and 5 are of no use, since the authors only used free software found on the internet, such as GenMANIA or STRING, to obtain information on the identified TRIM genes. These TRIM genes are however very well characterized and a bibliographic analysis would have been sufficient. In summary, the article attempts to hide the lack of results with an exaggerated and somewhat naive use of online algorithms.

Answer: We appreciate and respect the reviewer's comment. However, it's essential to consider that our approach, which involves reanalyzing already published RNA-seq datasets, is not new and has been widely used by several researchers. Many contributions to the field have been made through similar methodologies, and these studies have been published in various high-impact journals (1-3). Furthermore, as mentioned in the manuscript, most studies have focused on type I and II interferons, neglecting IFN-III. Additionally, these studies have typically utilized microarrays rather than RNA-Seq data to determine transcriptional patterns in response to treatment with the three types of IFNs, let alone IL-27.

To aid in understanding our methodological strategy better, we have added a diagram, which corresponds to Figure 1. We believe that our approach builds upon existing research and contributes valuable insights to the field. If you have any further questions or concerns, please feel free to let us know.

Specific comments

*Line 133: I don't think those are human macrophages in the article Rajsbaum et al. 2008. Data on human macrophages are from Martinez FO et al. J. Immunol. 2006

Answer: Thank you very much for the observation. We have already corrected that mistake.

*Line 202: Doses of interferon should be indicated in IU/ml. Concentrations in ng/ml mean nothing because it depends on the specific activity of interferons.

Answer: Following the reviewer's suggestion, the change was made to UI in the text

*Line 203: Which IFN-l ? Where are the references for IFN-a and -w ?

Answer: Unfortunately, we do not understand what the evaluator is asking, since in the manuscript, neither IFN-a nor much less IFN-w is mentioned. As written in the document, we refer to IFN-α and -ω, which have been widely described in the text, including Cartagena et al. The family of IFN-I comprises sixteen members in humans, including IFNβ, IFNε, IFNκ, IFNω, and 12 subtypes of IFNα (4).

*Figure 1A should include some positive and negative controls. But, once again, the data presented were not generated by the authors, with the exception of IL27.

Answer: Since we are using RNA-seq datasets available in the GEO, it's not possible to include controls in Figure 2 (previously 1).

*Table 1 is difficult to read and contains typos: "ngr/mL" instead of ng/mL, "Influeza" instead of "Influenza"

Answer: Thank you very much for the observation. Based on these suggestions, the Table 1 was modified.

References

  1. Keke Wu, Jiayi Zhu, Yingxu Ma, Yong Zhou, Qiuzhen Lin, Tao Tu, Qiming Liu. Exploring immune related gene signatures and mechanisms linking non alcoholic fatty liver disease to atrial fibrillation through transcriptome data analysis. Sci Rep. 2023 Oct 16;13(1):17548

  1. Fumie Horiuchi, Hiroshi Kumon, Rie Hosokawa, Kiwamu Nakachi, Kentaro Kawabe, Jun-Ichi Iga, Shu-Ichi Ueno. Identification of aberrant innate and adaptive immunity based on changes in global gene expression in the blood of adults with autism spectrum disorder. J Neuroinflammation. 2021 Apr 30;18(1):102

  1. Yue Li, Ye Liu, Mengjie Duo, Ruhao Wu, Tianci Jiang, Pengfei Li, Yu Wang, Zhe Cheng. Bioinformatic analysis and preliminary validation of potential therapeutic targets for COVID-19 infection in asthma patients. Cell Commun Signal. 2022 Dec 27;20(1):201.

  1. Piehler, J., Thomas, C., Garcia, K. C., & Schreiber, G. (2012). Structural and dynamic determinants of type I interferon receptor assembly and their functional interpretation. Immunological Reviews, 250(1), 317–334. https://doi.org/10.1111/imr.12001

Reviewer 2 Report

Comments and Suggestions for Authors

 In this study, the authors aim to address the gene regulation of TRIM family induced by Type-I,II,III interferons and IL-27 via bulk RNA sequencing approach. They highlight that the induction of TRIM genes is caused in monocyte-derived macrophages following treatment of interferons and IL-27, as evidenced by computational analysis. Additionally, they quantify the TRIM gene expression using qRT-PCR. Although overall the manuscript is well written, there are some major concerns to reach the standard of Viruses.

1. Major comments

* The authors performed the bulk RNA sequencing on macrophage cell lysates harvested at 18 hour post-treatment with interferons or IL-27. The result show that some TRIM genes were upregulated or downregulated by the treatment. To provide a comprehensive analysis, the authors should include other interferon-stimulated genes (ISGs) as controls in Fig. 1.

* The authors quantified mRNA levels of TRIM family using the macrophage cell lysates harvested at 24 hour post-treatment with interferons or IL-27. It is unclear why these quantified cell lysates were harvested differently from RNAseq samples. Additionally, since cell signaling by interferons or interleukins occurs rapidly, the authors should present mRNA levels via qRT-PCR in a time-dependent manner. ISG15, ISG56, MxA or OASL could serve as good controls.

* The authors should rewrite the result section 3.6. Validation of key TRIMs by RT-PCR.

2. Minor comments

* Line 255: correct typo error (/PML) to (PML).

* All figures need to be changed with better quality images and resize the images according to the journal’s format.

Author Response

Reviewer 2

Open Review

Quality of English Language

( ) I am not qualified to assess the quality of English in this paper
( ) English very difficult to understand/incomprehensible
( ) Extensive editing of English language required
( ) Moderate editing of English language required
( ) Minor editing of English language required
(x) English language fine. No issues detected

Yes

Can be improved

Must be improved

Not applicable

Does the introduction provide sufficient background and include all relevant references?

(x)

( )

( )

( )

Are all the cited references relevant to the research?

(x)

( )

( )

( )

Is the research design appropriate?

( )

( )

(x)

( )

Are the methods adequately described?

(x)

( )

( )

( )

Are the results clearly presented?

( )

( )

(x)

( )

Are the conclusions supported by the results?

( )

(x)

( )

( )

Comments and Suggestions for Authors

 In this study, the authors aim to address the gene regulation of TRIM family induced by Type-I,II,III interferons and IL-27 via bulk RNA sequencing approach. They highlight that the induction of TRIM genes is caused in monocyte-derived macrophages following treatment of interferons and IL-27, as evidenced by computational analysis. Additionally, they quantify the TRIM gene expression using qRT-PCR. Although overall the manuscript is well written, there are some major concerns to reach the standard of Viruses.

1. Major comments

* The authors performed the bulk RNA sequencing on macrophage cell lysates harvested at 18 hour post-treatment with interferons or IL-27. The result show that some TRIM genes were upregulated or downregulated by the treatment. To provide a comprehensive analysis, the authors should include other interferon-stimulated genes (ISGs) as controls in Fig. 1.

 Answer: Thank you for the suggestion. However, our study aimed to focus specifically on TRIMs for the comparative analysis of transcriptional profiles. We have already submitted another manuscript that covers both inflammatory and antiviral responses. Therefore, we focus on the present manuscript solely on TRIMs. We Include the figure with ISGs for informational purposes without incorporating them into the manuscript.

* The authors quantified mRNA levels of TRIM family using the macrophage cell lysates harvested at 24 hour post-treatment with interferons or IL-27. It is unclear why these quantified cell lysates were harvested differently from RNAseq samples. Additionally, since cell signaling by interferons or interleukins occurs rapidly, the authors should present mRNA levels via qRT-PCR in a time-dependent manner. ISG15, ISG56, MxA or OASL could serve as good controls.

Answer: We agree with the reviewer. However, the cell lysate was harvested at the same time as RNA-seq. This was our mistake and it was corrected.

* The authors should rewrite the result section 3.6. Validation of key TRIMs by RT-PCR.

Answer: We have rewritten the results section 3.6: Validation of key TRIMs by RT-qPCR.

  1. Minor comments

* Line 255: correct typo error (/PML) to (PML).

Answer: Its was corrected.

* All figures need to be changed with better quality images and resize the images according to the journal’s format.

Answer: All figures have been replaced with higher-quality images and resized according to the journal's format.

Round 2

Reviewer 1 Report

Comments and Suggestions for Authors

I realize that "reanalyzing already published RNA-seq datasets is not new and has been widely used by several researchers". However, in this particular case, I don't believe that this approach adds anything new to the discipline. It is well established which TRIM genes are induced by IFN in human and murine cells, and the function of most TRIM proteins is now known. The only new aspect of this manuscript is the induction of certain TRIM genes by IL27, but, as I wrote in my first review, this was unfortunately not followed by functional experiments to assess whether these TRIM proteins mediate the IL27-induced antiviral response.

The authors decided to ignore this major comment and not to carry out any further experiments. Under these conditions, I cannot recommend the acceptance of this manuscript.

Comments on the Quality of English Language

The English seems OK to me, but I'm not a native English speaker.

Author Response

Answer to comments of reviewer 2

I realize that "reanalyzing already published RNA-seq datasets is not new and has been widely used by several researchers". However, in this particular case, I don't believe that this approach adds anything new to the discipline. It is well established which TRIM genes are induced by IFN in human and murine cells, and the function of most TRIM proteins is now known. The only new aspect of this manuscript is the induction of certain TRIM genes by IL27, but, as I wrote in my first review, this was unfortunately not followed by functional experiments to assess whether these TRIM proteins mediate the IL27-induced antiviral response.

The authors decided to ignore this major comment and not to carry out any further experiments. Under these conditions, I cannot recommend the acceptance of this manuscript.

Answer: Following the reviewer comment, we decided to assess the effect of IL27 on Mayaro virus replication. To this end, we evaluated diverse concentrations of IL27. Furthermore, we quantified the mRNA level of various TRIMs. The findings of these experiments are depicted in Figure 8.

Reviewer 2 Report

Comments and Suggestions for Authors

I cannot accept the authors' response. The authors' answer is insuffient. The authors at least show the 18 hour RNA quntification data after treatment of interferon or IL-27.

Author Response

Answer to comments of reviewer 2

I cannot accept the authors' response. The authors' answer is insuffient. The authors at least show the 18 hour RNA quntification data after treatment of interferon or IL-27.

Answer: Following the reviewer comment, we decided to assess the effect of IL27 on Mayaro virus replication. To this end, we evaluated diverse concentrations of IL27. Furthermore, we quantified the mRNA level of various TRIMs. The findings of these experiments are depicted in Figure 8.

Furthermore, in the first evaluation, the reviews recommended quantified other ISGs. To answer this question, we have just published an article reporting the expression of these ISGs in response to treatment with IL27 and interferons (1)

References

  1. Interleukin 27, like interferons, activates JAK-STAT signaling and promotes pro-inflammatory and antiviral states that interfere with dengue and chikungunya viruses replication in human macrophages. Valdés-López JF, Hernández-Sarmiento LJ, Tamayo-Molina YS, Velilla-Hernández PA, Rodenhuis-Zybert IA, Urcuqui-Inchima S.Front Immunol. 2024 Apr 24;15:1385473. doi: 10.3389/fimmu.2024.1385473. eCollection 2024.